# Paternal multigenerational exposure to an obesogenic diet drives epigenetic predisposition to metabolic diseases in mice

Georges Raad[1,2†], Fabrizio Serra[1‡], Luc Martin[2], Marie-Alix Derieppe[2§],
Jérôme Gilleron[3], Vera L Costa[1], Didier F Pisani[2], Ez-Zoubir Amri[2],
Michele Trabucchi[1], Valerie Grandjean[1]*

[1]Université Côte d'Azur, Inserm, C3M, TeamControl of Gene Expression (10), Nice, France; [2]Université Côte d'Azur, CNRS, Inserm, iBV, Nice, France; [3]Université Côte d'Azur, Inserm, C3M, Team Cellular and Molecular Pathophysiology of Obesity and Diabetes (7), Nice, France

**Abstract** Obesity is a growing societal scourge. Recent studies have uncovered that paternal excessive weight induced by an unbalanced diet affects the metabolic health of offspring. These reports mainly employed single-generation male exposure. However, the consequences of multigenerational unbalanced diet feeding on the metabolic health of progeny remain largely unknown. Here, we show that maintaining paternal Western diet feeding for five consecutive generations in mice induces an enhancement in fat mass and related metabolic diseases over generations. Strikingly, chow-diet-fed progenies from these multigenerational Western-diet-fed males develop a 'healthy' overweight phenotype characterized by normal glucose metabolism and without fatty liver that persists for four subsequent generations. Mechanistically, sperm RNA microinjection experiments into zygotes suggest that sperm RNAs are sufficient for establishment but not for long-term maintenance of epigenetic inheritance of metabolic pathologies. Progressive and permanent metabolic deregulation induced by successive paternal Western-diet-fed generations may contribute to the worldwide epidemic of metabolic diseases.

*For correspondence:
grandjea@unice.fr

Present address: † Al-Hadi Laboratory and Medical Center, Beirut, Lebanon; ‡ Institutefor Maternal and Child Health IRCCS Burlo Garofolo, Trieste, Italy; § Universitéde Bordeaux, Bât B3, Allée Geoffroy St Hilaire, Pessac, France

Competing interests: The authors declare that no competing interests exist.

## Introduction

Nongenetic inheritance of newly acquired phenotypes is a relatively new concept in biology whereby changes induced by specific environmental cues in parents (mothers and/or fathers) can be transmitted to the next generations (*Chen et al., 2016*; *Fullston et al., 2013*; *Grandjean et al., 2016*). This process is evolutionarily conserved and has been described from worms to humans (*Gapp et al., 2014*; *Portha et al., 2019*; *Remy, 2010*; *Skinner et al., 2015*). The fact that environmental cues have the potential to modify the molecular hereditary information carried by the spermatozoa demonstrates that the environmentally induced epigenetic modifications (*Agarwal and Majzoub, 2017*) are not erased through the epigenetic reprogramming process, causing them to be inherited by the next generations (*Carone et al., 2010*; *Soubry, 2018*). Although the role of epigenetic modifications such as DNA methylation (*Carone et al., 2010*; *de Castro Barbosa et al., 2016*; *Ge et al., 2014*) and chromatin modifications (*Öst et al., 2014*; *Terashima et al., 2015*) cannot be excluded in this process, independent experimental data strongly evoke the central role of sperm RNA as a vector of paternal intergenerational epigenetic inheritance of, at least, environmentally induced metabolic pathologies (*Chen et al., 2016*; *Grandjean et al., 2016*; *Sharma et al., 2016*). Unlike genetic inheritance, environmentally induced epigenetic alterations are reversible, enabling the loss of previously

acquired characteristics (*Cropley et al., 2016*). Although environmental changes might persist over several generations, most reports have been based on the maintenance of paternal environmental cues for just one generation (*Huypens et al., 2016*). This is particularly true for certain lifestyle habits, such as eating high-fat or high-sugar junk food, also called a Western diet (WD). Thus, although people around the world may face multigenerational unbalanced nutrition, there have been limited studies on its effects on the metabolic health of the progeny.

Herein, we studied the impact of the paternal maintenance of an unhealthy WD for multiple generations on the metabolic phenotype of both the progenitors and their respective chow-diet-fed (CD-fed) offspring.

## Results

### Feeding successive paternal generations with a WD exacerbates the overweight phenotype and accelerates the development of obesity-associated pathologies

To test experimentally whether the maintenance of an unhealthy diet through the paternal germline influences the metabolic phenotype of the resulting individuals, C57BL6/J male mice were fed a WD for five consecutive generations (from WD1 to WD5) (*Figure 1A*, *Figure 1—figure supplement 1A*). According to a previous study (*Massiera et al., 2010*), multigenerational WD feeding exacerbates the increased body weight mass induced by this diet. Despite marked heterogeneity in the WD4 and WD5 populations, we found that the WD4 and WD5 males weighted significantly more than the WD1 and WD3 ones (p<0.05 and <0.01, respectively) (*Figure 1B, Figure 1—figure supplement 2A, B*). Interestingly, growing heterogeneity of the body weight mass between the males of the first and the latter generations (*Figure 1—figure supplement 1C*) was observed in the four independent families, indicating that the phenotypic heterogeneity previously observed in diet-induced obesity mouse models (*Burcelin et al., 2002*) increases progressively over the generations. This increase in total body weight with paternal multigenerational WD feeding was associated with an increase in perigonadal white adipose tissue (gWAT) mass (*Figure 1C, Figure 1—source data 1*). Indeed, WD5 gWAT weighted significantly more than the WD1 gWAT (p<0.05) and the gWAT volume measured by computed tomography increased 2.3-fold and 3.4-fold in WD1 and WD5 mice, respectively, compared to that of control mice (CD-fed mice) (*Figure 1—source data 1*). The increase in gWAT mass was positively correlated with total body weight (perigonadal fat mass versus total body weight; Spearman's *r* = 0.78, p<0.0001) (*Figure 1—figure supplement 2E*). It was also associated with the hypertrophy of white adipocytes, with a median surface cell area of white adipocytes increasing from 1500 to 4000 $\mu m^2$ from the first (WD1) to the fifth generation (WD5) and with a decreased calculated number of adipocytes in WD5 compared to the controls (*Figure 1D–F*). Furthermore, our RNA-seq comparison between the gWAT of WD1 and WD5 males revealed that multigenerational WD feeding has a strong impact on the gWAT gene expression profile. In fact, we observed an increase in differentially expressed genes (DEGs), from 325 in WD1 (with 93 upregulated and 232 downregulated genes) to 1199 (757 upregulated and 442 downregulated) in WD5, compared to the respective CD-fed mice. Interestingly, while the majority of DEGs in WD1 (66%) were also deregulated in WD5, a minority of DEGs in WD5 (only 8% for the upregulated genes and 35% for the downregulated genes) were deregulated in WD1 (p-value<0.01). Importantly, all common genes were deregulated in the same direction (*Figure 1G*). Interestingly, querying the WD1 and WD5 DEGs against the molecular signature database collection of curated gene pathway annotations revealed a specific WD5 enrichment in gene sets associated with CHEN_METABOLIC_SYNDROM_NETWORK (genes forming the macrophage-enriched metabolic network claimed to have a causal relationship with metabolic syndrome traits) and with genes potentially regulated by the methylation of lysine 4 (H3K4) and lysine 27 (H3K27) of histone H3 and by polycomb repressive complex 2 (PRC2) (*Figure 1—source data 2*; *Liberzon, 2014*).

The aforementioned modulations of white adipose tissue in WD generations shed light on the possible exacerbation of obesity-associated pathologies (such as insulin resistance [and subsequently type II diabetes] and nonalcoholic fatty liver disease) (*Gilleron et al., 2018*). To check this hypothesis, several metabolic risk parameters related to these pathologies were analyzed in WD-fed mice (*Table 1*). In comparison with CD-fed mice, circulating plasma levels of leptin, C-reactive protein

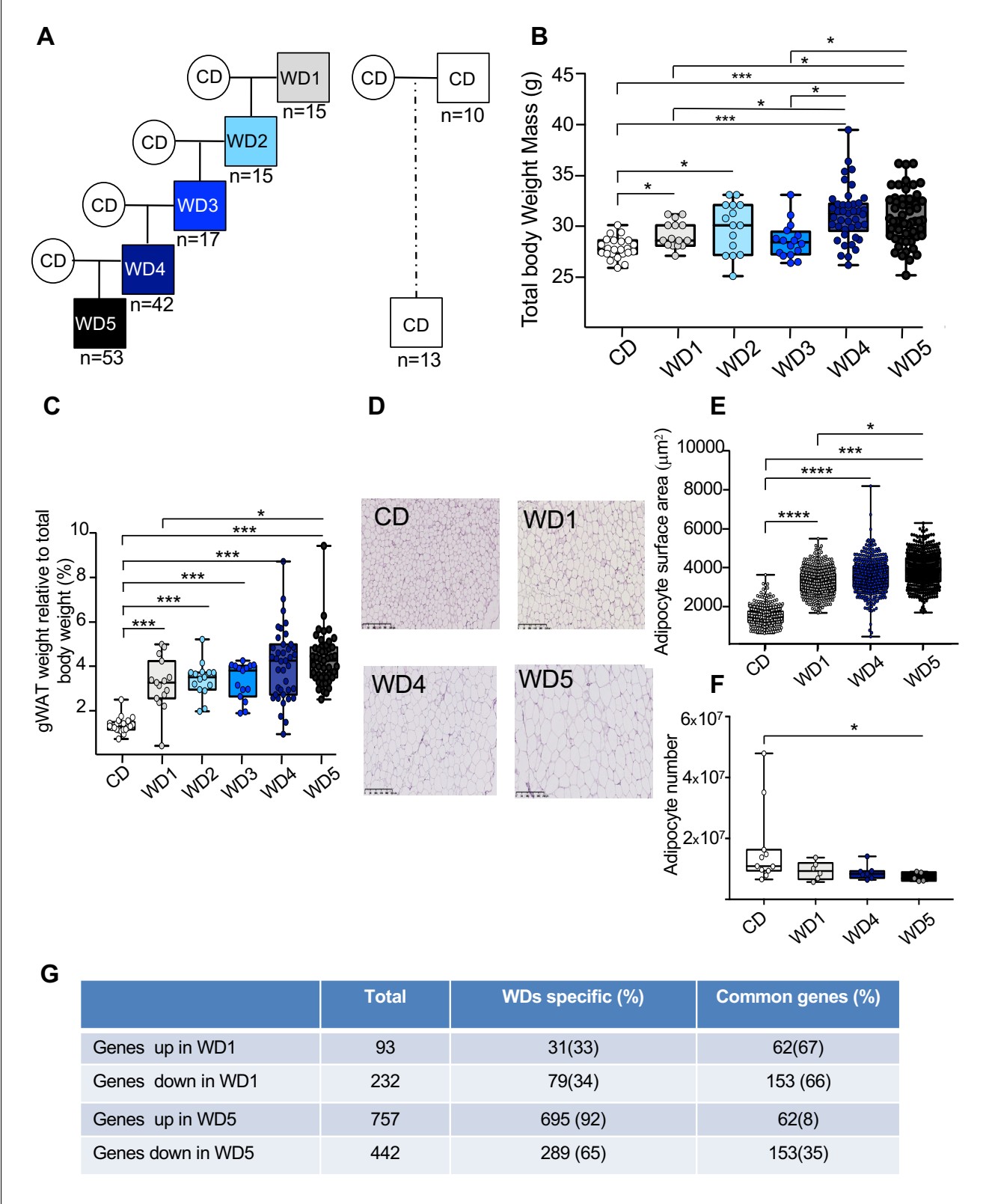

**Figure 1.** Five consecutive paternal generations of Western diet (WD) feeding exacerbate the WD-induced overweight phenotype. (**A**) Study design for the maintenance of WD feeding for five consecutive generations through the paternal lineage. Male mice were randomized to receive either a control diet (CD; 5% of energy from fat) or a WD (WD1; 45% of energy from fat). After 3 months of WD feeding, four males obtained from different fathers were arbitrary chosen to generate four independent families. They were mated with CD-fed females to generate WD2 offspring. At 3 weeks old, they were

*Figure 1 continued on next page*

*Figure 1 continued*

fed a WD, and at 4 months, at least one WD-fed male per family was crossed with CD-fed females. This second generation of males was called WD2. This experimental design was repeated three times to obtain the WD5 group (*Figure 1—figure supplement 1*). (B) Box-whiskers (min-max) of the median total body weight of the different male WD cohorts (n ≥ 8 mice per group). (C) Box-whiskers (min-max) of the median perigonadal white adipose tissue (gWAT) weight relative to total body weight in the different WD cohorts. (D) H&E staining of gWAT sections (scale bar: 200 µm) in representative CD, WD1, WD4, and WD5 males. (E) Box-whiskers (min-max) of the median surface area (µm$^2$) of the adipocytes, which was calculated using Image Analyzer software (ImageJ). The total count ranged from 3275 to 7052 cells per condition (n ≥ 4 mice per group). (F) Box-whiskers (min-max) of the number of adipocytes, which was estimated using the mathematical equation developed by *Jo et al., 2009*, as previously described in *Gilleron et al., 2018*. (G) Table showing the differentially expressed genes in WD1 and WD5 perigonadal white adipose tissue. *$p_{adj}$<0.05, **$p_{adj}$<0.01, ***$p_{adj}$<0.001, ****$p_{adj}$<0.0001.

The online version of this article includes the following source data and figure supplement(s) for figure 1:

**Source data 1.** Physiological characteristics of different WD groups.
**Source data 2.** Molecular signature database collection-Curated gene set enrichment from analyzing the differentially regulated genes in WD1 and WD5 males.
**Figure supplement 1.** Schematic representation of the experimental procedure.
**Figure supplement 2.** Exacerbation of the overweight phenotype upon continuous paternal Western diet (WD) feeding for multiple generations.

(CRP), one marker of inflammation, and total cholesterol were significantly higher in the WD3 (p<0.01), WD4 (p<0.05), and WD5 (p<0.01) groups but not in the WD1 (p=0.07) or WD2 (p=0.4) groups (*Table 1*). The alterations in these metabolic parameters in WD-fed males were found to be positively correlated with the increase in gWAT mass (*Figure 1—figure supplement 2F–H*). At the molecular level, the increase in serum leptin in WD-fed males was positively correlated with an increase in leptin mRNA levels in the gWAT of the respective male mice (total serum leptin and *leptin* mRNA, Spearman's *r* = 0.89, p<0.0001, *Figure 1—figure supplement 2I*), suggesting an accumulation of epigenetic modifications at the leptin promoter. These results are in line with recent studies showing that leptin upregulation occurs via epigenetic malprogramming in white adipose tissue (*Lecoutre et al., 2017*; *Masuyama et al., 2016*). Furthermore, we found a significantly impaired response in the intraperitoneal glucose tolerance test (GTT) in all WD-fed mouse groups (*Figure 2A*), which was not associated, except for in WD2-fed males, with an impaired insulin response, as shown by the intraperitoneal insulin tolerance test (ITT) (*Figure 2B*). Therefore, unlike the other metabolic parameters, we did not notice any significant exacerbation of insulin sensitivity in successive generations. Moreover, the response to an intraperitoneal GTT (measured through the area under the curve [AUC]-GTT calculation) was not correlated with the gWAT mass (*Figure 1—figure supplement 2J*). Together, these data might reflect the multifactorial and complex nature of the pathogenesis of obesity-induced diabetes.

Strikingly, although the C57BL6/J-strain male mice fed a WD diet for one generation failed to develop strong alterations in liver phenotype (*Schierwagen et al., 2015*; *AMDCC et al., 2005*), major abnormalities were observed in WD5 liver, that is, organ weight, histological and biochemical parameters. Indeed, the mass of the WD5 liver (not that of the WD1 liver) was significantly higher than that of the CD specimens (*Figure 2C*). Furthermore, unlike WD1 liver, histological and biochemical examinations revealed the presence of macrovesicular steatosis with significantly increased

**Table 1.** Evolution of serum biomarker parameters in different WD groups.

| Parameters | Control n = 6 | WD1 n = 4 | WD2 n = 5 | WD3 n = 7 | WD4 n = 7 | WD5 n = 7 |
|---|---|---|---|---|---|---|
| Adiponectin (ng/ml) | 4.4 (4.9–5.2) | 5.2 (3.3–6.9) | 3.5 (2.8–4.7) | 3.4 (2.8–5.5) | 3.6 (2.9–4.2) | 4.6 (3.5–5.0) |
| Leptin (ng/ml) | 6.7 (5.1–7.5) | 10.2 (6.2–11) | 8.2 (6.2–11.6) | 11.5 (7.9–28) | **19 (13–32)**\*\*[2] | **21 (12–26)**\*\*[2] |
| CRP (g/ml) | 4.2 (3.2–5.0) | 5.8 (5.1–6.2) | 5.7 (5.4–6.2) | **7.1 (5.7–9.8)**\*\* | **5.7 (3.5–7.9)**\* | **5.8 (4.6–7.4)**\*\* |
| Total cholesterol (mg/dl) | 1.1 (0.9–1.2) | **1.8 (1.3–2.0)**\*\* | 1.4 (1.0–1.6) | **1.6 (1.5–1.9)**\* | **1.9 (1.6–2.4)**\*\*\*[2] | **1.9 (1.6–2.4)**\*\*\*[2] |

Values are expressed as median (IQR). Numbers are in bold if $p_{adj}$<0.05.

\* and 2 denote the WD groups significantly different from that of the CD and WD2 groups, respectively.

\*$p_{adj}$<0.05, \*\*$p_{adj}$<0.01.

WD: Western diet; CD: control diet; CRP: C-reactive protein.

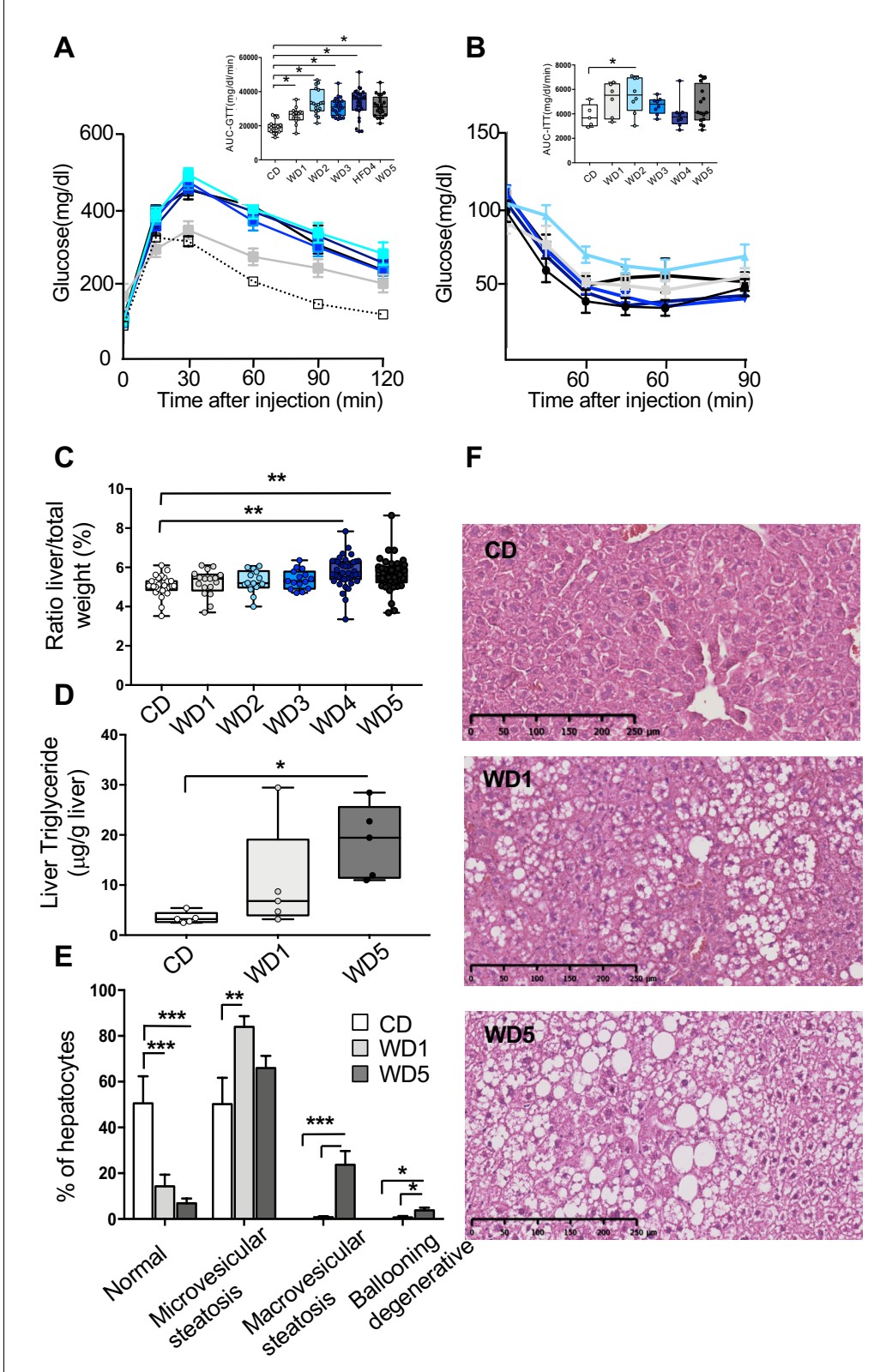

**Figure 2.** Five consecutive paternal generations of Western diet (WD) feeding exacerbate WD-induced overweight pathologies. (A, B) Evolution of glucose parameters in male mice fed a WD for five successive generations. Blood glucose and insulin tolerance tests were performed on 16-week-old males (n ≥ 6). Plasma glucose (inserted box-whiskers [min-max] of the median area under the curve [AUC] and above baseline for glucose from time point 0 to 120; glucose tolerance test) (A); (inserted box-whiskers [min-max] of the median AUC and above baseline for glucose from time point 0 to

*Figure 2 continued on next page*

*Figure 2 continued*

100; insulin tolerance test) (**B**). Glucose tolerance and insulin tolerance tests were conducted in the morning in overnight-fasted mice. (**C**) Box-whiskers (min-max) of the median liver weight relative to total body weight in the different WD cohorts (n ≥ 8 mice per group). (**D**) Liver triglyceride contents in the control diet (CD), WD1, and WD5 groups (n ≥ 6). (**E**) Percentage of normal hepatocytes, hepatocytes with microvesicular steatosis, hepatocytes with macrovesicular steatosis, and ballooning degenerative hepatocytes in CD, WD1, and WD5 livers (n ≥ 6). (**F**) H&E staining of liver sections (scale bar: 250 µm) from representative CD, WD1, and WD5 males. *$p_{adj}$<0.05, **$p_{adj}$<0.01, ***$p_{adj}$<0.001.

triglyceride (TG) levels in WD5 liver compared with CD liver (p<0.01, respectively) (*Figure 2D–F*). Therefore, the phenotype of WD5 livers exhibits typical features of fatty liver.

Together, both morphological and molecular features demonstrate that multigenerational WD feeding induced a progressive dysregulation of the male metabolic phenotypes (*Figure 1—figure supplement 2K*), with an exacerbation of the gWAT size and gWAT transcriptional alteration as well as of obesity-associated pathologies such as fatty liver. Therefore, a worsening of the underlying medical conditions can be potentially transmitted to next generations.

## Long-term transgenerational epigenetic inheritance of an overweight 'healthy' phenotype

Previous reports showed that WD-induced metabolic dysregulations during one-generation exposure could be transmitted across one (F1) or two generations (F2) fed a CD (*Fullston et al., 2013*; *de Castro Barbosa et al., 2016*). To investigate the impact of feeding a WD through several generations on the inheritance of diet-induced metabolic pathologies, we compared the metabolic status of F1, F2, and F3 cohorts fed a CD generated from either CD, WD1, or WD5 males (*Figure 3A*). As expected from previous studies (*Fullston et al., 2013*; *Grandjean et al., 2016*), male and female F1 progenies derived from WD1 males (F1-WD1) were statistically heavier than the control animals with CD-fed ancestors (*Figure 3B, F*, *Figure 3—figure supplement 1A–C*). Although the difference did not reach significance at the age of 16 weeks, the same trend was also observed for the F1 progenies derived from WD5 (F1-WD5) male progenies (*Figure 3D, H, Figure 3—figure supplement 1A–C*). This overweight phenotype was associated with impaired glucose tolerance as measured by the GTT for both the male F1-WD1 and F1-WD5 progenies and the female F1-WD5 mice (*Figure 3—figure supplement 1E, G, Figure 3—source data 1* and *2*). We noticed, however, the absence of intergenerational inheritance of the fatty liver phenotype observed in the WD1 and WD5 progenitors (*Figure 3—figure supplement 2*).

Both male F2-WD1 and F2-WD5 CD-fed progenies were also overweight (p<0.01). This phenotype was associated with an excessive accruement of gWAT mass of at least 90% over the control (*Figure 3C, E, G, I*). Importantly, although the female and male F2-WD5 progenies were found to be significantly fatter and heavier than the F2-WD1 cohorts, these mice did not exhibit impaired glucose tolerance (as measured by the GTT) (*Figure 3—figure supplement 1E, G*) or signs of fatty liver lesions (*Figure 3—figure supplement 2*).

The metabolic differences were even more striking in both F3 and F4 progenies (*Figure 3*, *Figure 3—figure supplement 1*). Thus, as illustrated in *Figure 3B–I*, the populations of males and females of the F3-WD1 progenies were very homogeneous, exhibiting metabolic characteristics very similar to control mice. By contrast, both populations of males and females of the F3-WD5 progenies were heterogeneous in terms of body and gWAT weights, some of them showing weights closed to CD mice and others being clearly overweight and fat. However, both F3-WD5 populations were significantly heavier and fatter (p<0.001 and <0.01, respectively) than control and F3-WD1 populations (*Figure 3B–I, Figure 3—source data 1* and *2*). Strikingly, despite being overweight, the progenies derived from WD5-fed animals did not display any alterations in terms of glucose metabolism (*Figure 3—figure supplement 1E–H*) and fatty liver pathologies at 4 months of age (*Figure 3—figure supplement 2*, *Figure 3—source data 1* and *2*).

Collectively, these data suggest that WD feeding for multiple generations induces stable germline epigenetic modifications that were not erased after removing the stressor(s) for at least four generations of CD-fed progeny.

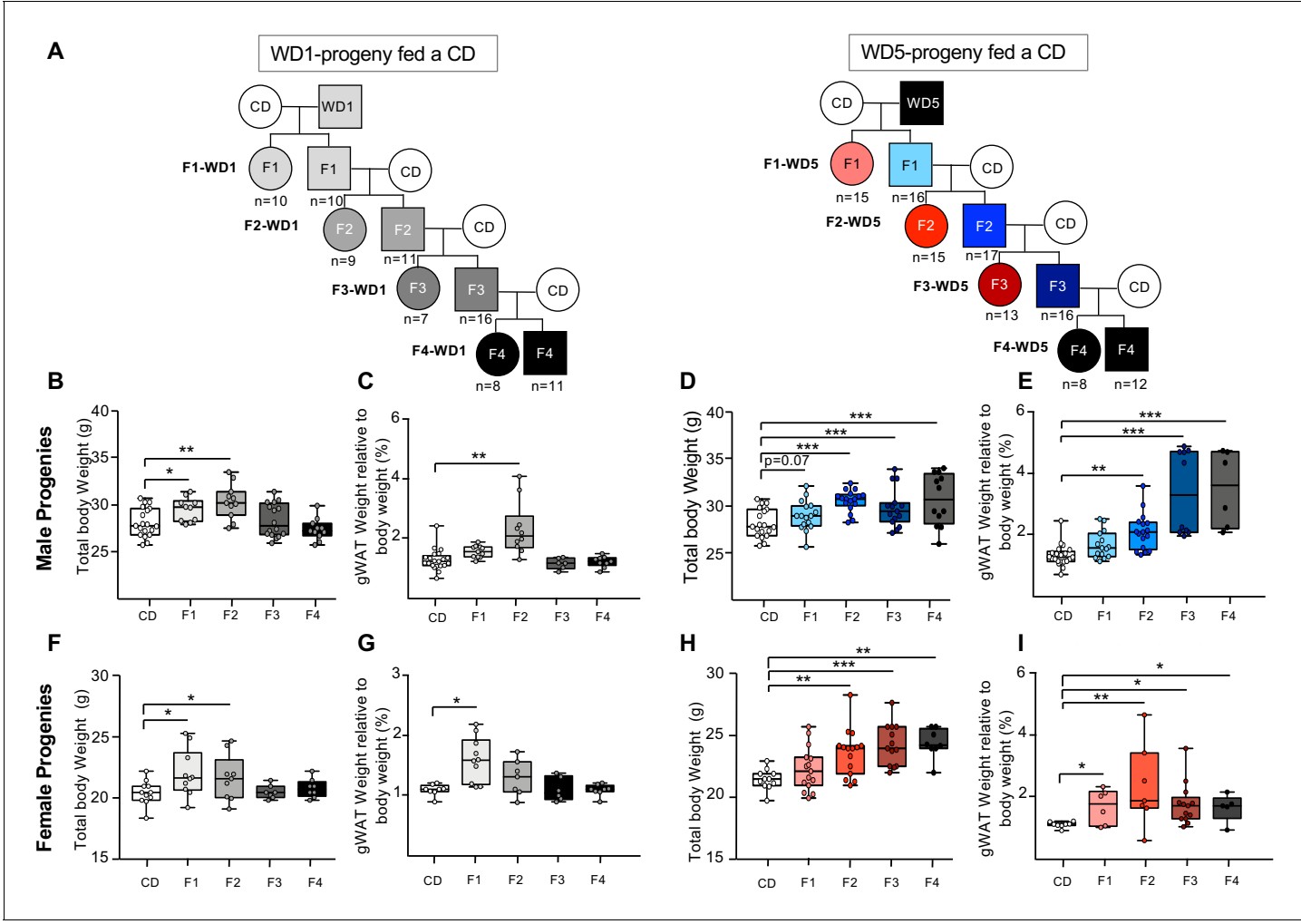

**Figure 3.** Maintenance of the overweight phenotype after four generations on the control diet (CD) in the progenies generated from Western diet (WD) 5-fed males. (**A**) Study design for the inheritance of WD-induced metabolic alterations in WD1- and WD5-fed animals. Four WD1 and nine WD5 male mice from different litters (arbitrary selected from the four different families) were mated with CD-fed females to generate F1-WD1 and F1-WD5 offspring, respectively. Each offspring was fed the CD. This crossing scheme was repeated three times to obtain the F2-, F3-, F4-WD1 and F2-, F3-, F4-WD5 offspring. The number of mice is indicated. Box-whiskers (min-max) of the median total body weights of 16-week-old males (**B, D**) and females (**F, H**) of progenies from WD-fed animals. Box-whiskers (min-max) of the median perigonadal white adipose tissue (gWAT) of males (**C, E**) and females (**G, I**) of progenies from WD-fed animals. The unimodality/multimodality of distributions for body weight for all groups was tested using the Hartigan's Dip Test for unimodality/multimodality. All groups, except the F1-WD1 male progenies, followed a unimodal distribution. Gray rectangles and circles represent the male and female progenies, respectively, from WD1-fed animals. Blue rectangles and red circles represent the male and female progenies, respectively, from WD5-fed animals. *$p_{adj}$<0.05, **$p_{adj}$<0.01, ***$p_{adj}$<0.001.

The online version of this article includes the following source data and figure supplement(s) for figure 3:

**Source data 1.** Physiological characteristics of F1, F2 and F3 male progenies from either WD1 or WD5 males.

**Source data 2.** Physiological characteristics of F1, F2 and F3 female progenies from either WD1 or WD5 males.

**Figure supplement 1.** Long-term epigenetic inheritance of a 'healthy' overweight phenotype in control diet (CD)-fed progenies from Western diet (WD)5 males.

**Figure supplement 2.** No liver alteration was observed in the control diet (CD)-fed progenies from Western diet (WD)1 and WD5 males.

## Sperm RNAs transmit only transient epigenetic inheritance of WD-induced pathologies

Specific signatures of sperm small non-coding RNAs (sncRNAs) from WD-fed mice have been previously shown to act as a vector of intergenerational epigenetic inheritance of newly acquired pathologies (*Chen et al., 2016*; *Grandjean et al., 2016*; *Sharma et al., 2016*; *Sarker et al., 2019*). To determine whether sperm small RNAs are also involved in the long-term maintenance of epigenetic

inheritance (transgenerational epigenetic inheritance), we systematically profiled the expression of sncRNAs of three independent CD, WD1, and WD5 sperm samples by deep sequencing using a recently developed sncRNA annotation pipeline optimized for rRNA- and tRNA-derived sRNAs, rsRNA, and tsRNA, respectively (*SPORTS1.0*) (*Shi et al., 2018*). We found a sperm RNA signature that is specifically induced in the first generation of WD-diet males characterized by an increase in the proportion of rsRNA. This is consistent with previous reports showing the sensitivity of rsRNA to dietary exposure such as a high-fat diet (*Nätt et al., 2019*; *Zhang et al., 2018*; *Figure 4B*). Strikingly, this sncRNA signature observed in the first generation of WD-fed male fed a WD tends to disappear in the fifth generation of WD-fed male, which exhibit a sncRNAs signature very similar to that of CD sncRNAs. On the one hand, while ~64% of the sncRNAs were annotated to rsRNAs in WD1, only ~40% of them were annotated to rsRNAs in CD and WD5 sperm (*Figure 4D*), suggesting a global modification of the sncRNA population upon WD exposure. On the other hand, a closer look at the sncRNAs differentially expressed in WDs (WD1 and WD5) and CD showed that the rsRNAs are not the only population to be sensitive to dietary exposure (*Figure 4*), which is consistent with previous reports in mice. Indeed, searching for small RNA DEGs (adjusted p value<0.05) between WD (WD1 or WD5) and CD sperm, we identified 479 and 66 DEGs in WD1 and WD5, respectively, compared to control mice (*Figure 4E, Figure 4—source data 1* and *2*). Interestingly, the majority of WD5 DEGs (47 out of 66) were already deregulated in WD1 (*Figure 4E*). Among these common sncRNAs, we identified tsRNA and microRNAs known to be involved in short-term epigenetic inheritance of metabolic dysfunction (intergenerational inheritance) (*Chen et al., 2016*; *Grandjean et al., 2016*; *Figure 4F, Figure 4—source data 1* and *2*). Interestingly, no sncRNA was found deregulated (adjusted p<0.05) between WD1 and WD5. These data indicate that the sensitivity of sperm sncRNA signature to diet, observed independently by several groups (*Nätt et al., 2019*; *Zhang et al., 2018*), is also modulated by the diet of ancestors.

To further investigate the role of sperm RNAs in the long-term transgenerational epigenetic inheritance of metabolic alterations, microinjection experiments into naive zygotes were performed with total sperm RNA from either WD1 or WD5 males (RNA-WD1 progenies and RNA-WD5 progenies, respectively) (*Figure 5A*). As previously reported, this experiment faithfully reproduces the pattern of short-term paternal transmission of environmentally induced phenotypes in crosses (*Chen et al., 2016*; *Grandjean et al., 2016*; *Gapp et al., 2014*; *Sharma et al., 2016*; *Sarker et al., 2019*). In agreement with previous studies, male 12-week F1-RNA-WD1 and F1-RNA-WD5 progenies were heavier than F1-RNA-CD progenies (31 g vs. 30 g, p<0.05) (*Figure 5B*). In addition, they displayed glucose and insulin response alterations, as shown by GTT and ITT analyses, with significantly higher values of the AUC to controls (*Figure 5D, E, Figure 5—source data 1*). Regarding the fatty liver phenotype, neither abnormal TG levels nor histological abnormalities were observed in liver from F1-RNA-CD and F1-RNA-WD progenies. Thus, the metabolic alterations observed in F1-RNA progenies are partially reminiscent of the WD1 and WD5 male phenotype.

Overweight phenotypes and glucose response alterations were partially transmitted to the F2 and F3 generations (*Figure 5—source data 2* and *3*). Indeed, total body weight mass of the F2-RNA-WD males was significantly heavier than that of the F2-RNA-CD male (p<0.05). This metabolic abnormality did not persist in the F3 and F4 progenies. Strikingly, although the glucose and insulin response alterations were not observed in the F2 generations of RNA-WD males, we noticed an alteration in those responses in the F3 generations of the RNA-WD5 males but not of the RNA-WD1 males. Intriguingly, while we did not observe any liver abnormalities in F1-RNA progenies, liver histological examinations revealed macro- and microvesicular steatosis in hepatocytes of two F2-WD overweight males (2 out of 10) (*Figure 5—figure supplement 1*). It should be noted that these abnormal hepatocytes were never observed in RNA-CD progenies. Nevertheless, all the metabolic alterations were completely absent in the F4 generations (*Figure 5—source data 4*).

Thus, the metabolic observed phenotype of WD1 and WD5 progenies obtained by either RNA microinjection or natural mating exhibited some discrepancies. First, the overweight phenotype induced by sperm RNA from WD5 males was not exacerbated compared to that induced by sperm RNA from WD1 males. In fact, no statistically significant difference was observed among the body weight of the F1, F2, and F3 progenies derived from sperm RNA of WD1- and WD5 animals. Second, the sperm-RNA-induced overweight phenotype was associated with glucose metabolic alterations (total body weight and AUC-GTT, Spearman's *r* = 0.4, p<0.01, *Figure 5F*) and was sporadically associated with fatty liver abnormalities in both WD1 and WD5 (*Figure 5—figure supplement 1*).

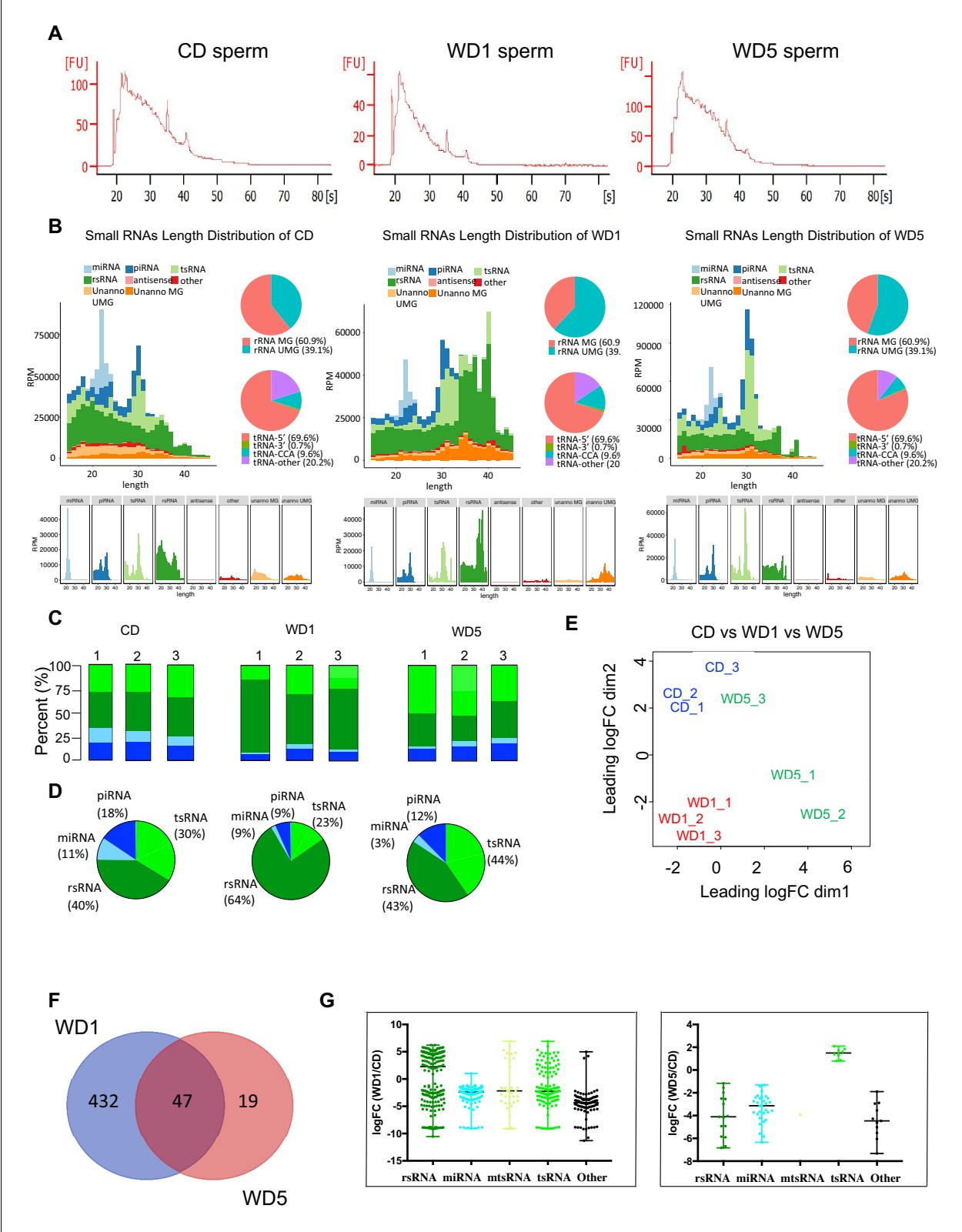

**Figure 4.** Sperm small non-coding RNAs (sncRNAs) signature modulated by the ancestors' diet. (**A**) Representative bioanalyzer profiles of control diet (CD), Western diet (WD)1, and WD5 sperm total RNAs. (**B**) Length distribution and pattern changes of sperm sncRNA different populations (miRNAs piRNAs, tsRNAs, and rsRNA) in one representative CD, WD1, and WD5. Each graph was generated by SPORTS1.0 (27). (**C**) Small RNA-seq profiles of each CD, WD1, and WD5 male. (**D**) Mean proportion of each small RNA population across each group. (**E**) The normalized small RNA levels

*Figure 4 continued on next page*

*Figure 4 continued*

from the CD (blue spots), WD1 (red spots), and WD5 (green spots) sperm were analyzed by principal component analysis (PCA). One WD5 fell outside the PCA cluster and was arbitrarily removed for differential expression analysis. (**F**) Venn diagram of small RNA sequences differentially expressed in WD1 and WD5 sperm. The numbers of small RNAs that are unique for each WD1 and WD5 male are shown in each circle. The numbers of genes in overlapping (common) are indicated at the intersections of the sets in the Venn diagram ($P_{adj}$<0.05 Log2FC≥|0.6|). (**G**) Scatter plots of microRNAs or piRNAs, tRNA fragments, and other small RNAs differentially expressed ($P_{adj}$<0.05 |Log2FC| ≥ 0.6) in WD1 (left panel) and WD5 (right panel) sperm compared to their expression in the CD sperm cohort.

The online version of this article includes the following source data for figure 4:

**Source data 1.** Differentially expressed small RNAs in WD1 sperm.
**Source data 2.** Differentially expressed small RNAs in WD5 sperm.

Taken together, these data strongly suggest that sperm RNAs can unequivocally induce intergenerational phenotype but may induce some transgenerational features, although the effect is weaker than the effect induced by whole sperm. This is in line with previous study using a psychological-stress-induced model showing that sperm RNA injection can also induce phenotype in both F1 and F2 generations (*Gapp et al., 2014*). Last, in contrast to natural mating of WD5, the sperm-RNA-induced overweight phenotype was not inherited for more than two generations. Together, these results indicate that sperm RNAs are not sufficient for the long-term epigenetic inheritance of metabolic dysfunctions.

## Discussion

Growing evidence suggests that an unbalanced diet of the father negatively affects its metabolic health and that of its progenies. Of particular interest, little attention has been focused on the effect of paternal successive generations of unbalanced diet exposure on metabolic health, which may have public health and economic impacts. To this end, we fed male mice for five successive generations on a high-fat, high-sugar diet (WD) to compare the metabolic parameters across multiple generations of WD males and assess the persistence of the WD-induced metabolic alterations in their subsequent balanced CD-fed progenies.

In summary, our findings reveal that maintaining a WD for several generations promotes a progressive accumulation of epigenetic alterations in somatic and germ cells throughout generations. Two lines of evidence support this conclusion. First, ancestral exposure influences the magnitude of the overweight phenotype. Indeed, a male whose father, grandfather, great grandfather, great-great-grandfather, and great-great-great grandfather, up to five generations of exposure, have been fed a WD exhibits the most severe overweight phenotype associated with serious metabolic alterations. Second, the father's ancestral history (whether his ancestors were fed an unbalanced diet) affected the pattern of inheritance of this metabolic pathology. Second, the father's ancestral history (whether or not the ancestors were fed an unbalanced diet) affected the pattern of inheritance of metabolic pathologies.

The main limitation of our study is the phenotypic heterogeneity observed in the males of the WD4 and WD5 generations (*Figure 1—figure supplement 1C*) and in the CD-fed WD5 progenies (*Figure 3*), which might lead to biased conclusions. Indeed, although the statistical tests we used here should rule out this weakness, we cannot rule out the possibility of the presence of subpopulations. Metabolic heterogeneity induced by an unbalanced diet has already been reported in mice (*Burcelin et al., 2002*; *Dumas et al., 2017*), and the strong heterogeneity observed in our model may indicate an adaptative process whereby different subpopulations could emerge in response to the maintenance of an unbalanced diet.

Although it is well described that the development of type 2 diabetes is positively associated with body weight (*Golay and Ybarra, 2005*), we did not observe a strong correlation between fat mass and glucose and insulin sensitivities in males obtained after multigenerational WD feeding. However, we identified one obese-associated pathology that increased in severity with successive generations of a WD, namely, hepatic steatosis, indicating that exposure sensitivity is heightened by multiple generations of exposure, at least for this diet-associated pathology. Thus, the family food

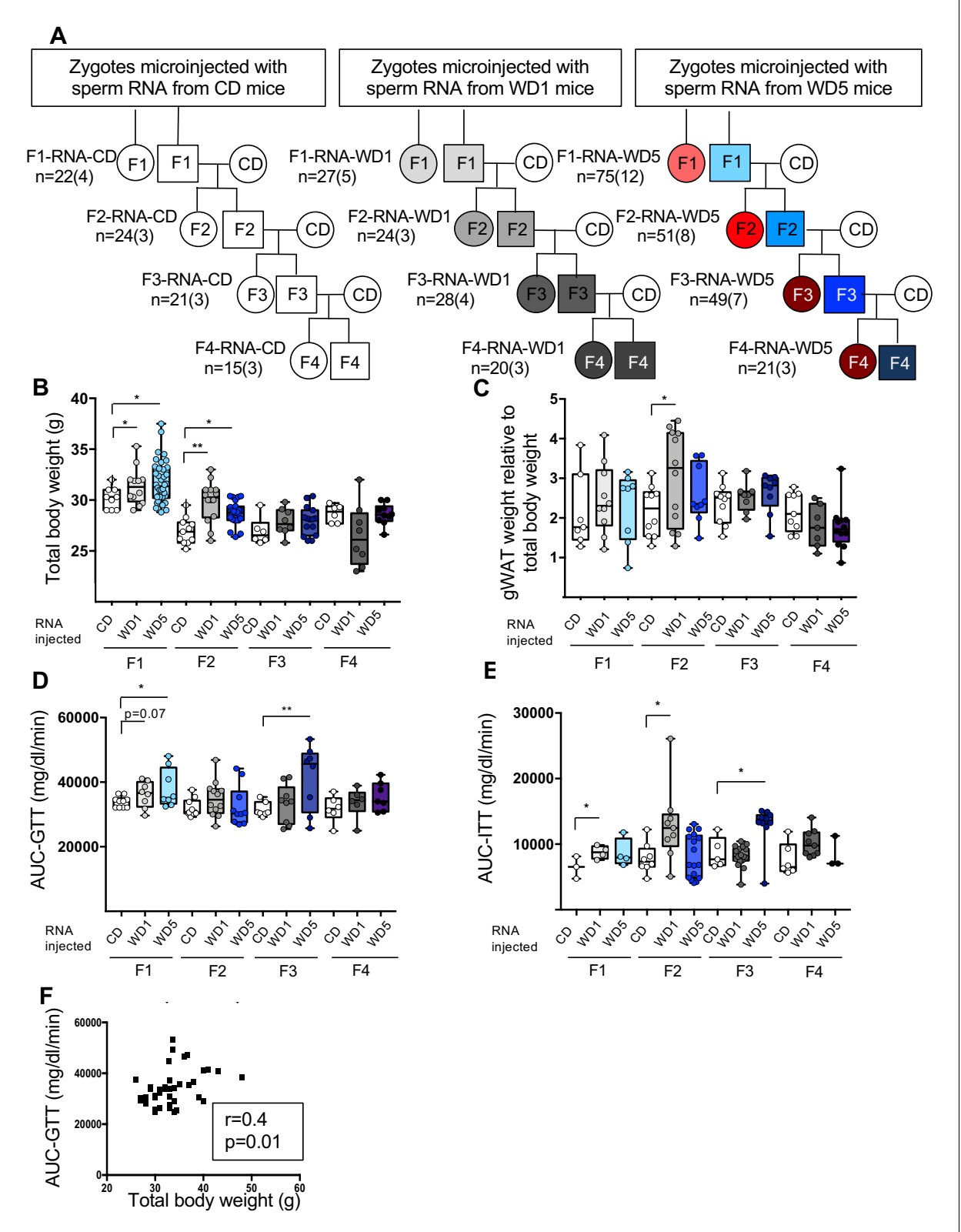

**Figure 5.** Zygotic microinjection of sperm total RNA from either Western diet (WD)1 or WD5 males induces metabolic alterations in the F1 and F2 control diet (CD)-fed progenies that are not maintained in the F3 and F4 CD-fed progenies. (**A**) Study design for the inheritance of metabolic alterations induced after the microinjection of sperm total RNA from CD-, WD1-, or WD5-fed males into C57BL/6J zygotes. Five F1 CD-fed males from each set of RNA microinjections were mated with CD-fed females to generate F2-RNA offspring. Each offspring was fed a control diet. This crossing

*Figure 5 continued on next page*

*Figure 5 continued*

scheme was repeated twice to obtain the F3-RNA offspring and then the F4-RNA offspring. (B) Box-whiskers (min-max) of the median total body weight of the F1-, F2-, F3-, and F4-RNA male progenies (n ≥ 8 mice per group). (C) Box-whiskers (min-max) of the median perigonadal white adipose tissue (gWAT) weight relative to total body weight in the different RNA progenies. The evolution of glucose parameters in male mice from RNA-injected progenies. (D) Box-whiskers (min-max) of the median area under the curve–glucose tolerance test (AUC-GTT) of each cohort. (E) Box-whiskers (min-max) of the median AUC-ITT of each group. (F) Bivariate correlation between the body weight of the F2-RNA-CD, F2-RNA-WD1, and F2-RNA-WD5 progenies and the AUC-GTT (n = 38). This correlation was similar using parametric (Pearson, $r = 0.4$, p=0.01) or nonparametric (Spearman, $r = 0.4$, p=0.01) correlations. *$p_{adj}$<0.05, **$p_{adj}$<0.01, ***$p_{adj}$<0.001.

The online version of this article includes the following source data and figure supplement(s) for figure 5:

**Source data 1.** Physiological characteristics of male and female F1-RNA-progenies.

**Source data 2.** Physiological characteristics of male and female F2-RNA- progenies.

**Source data 3.** Physiological characteristics of male and female F3-RNA- progenies.

**Source data 4.** Physiological characteristics of male and female F4-RNA- progenies.

**Figure supplement 1.** Sperm RNA microinjection did not fully recapitulate the fatty liver phenotype of the WD5 father.

environment, parental dietary behaviors, and family obesity might be an additional clue to explain the increasing incidence of nonalcoholic fatty liver disease in humans (*Kumar et al., 2020*).

Importantly, multiple generations of WD exposure impact not only the sensitivity to a WD but also the hereditary makeup, also called background. Indeed, when the father has no WD-fed ancestor, the fatness of its progenies tends to disappear after WD removal. However, in the case of fathers with several WD-fed ancestors, the progenies will remain stably overweight for more than four generations. Intriguingly, although the male progenies of the third and fourth generations of WD5 males were overweight, they did not develop metabolic alterations, such as glucose/insulin sensitivity alterations and fatty liver diseases. Together, these data strongly suggest that the combination of ancestral and individual diet exposure was both necessary and sufficient to elicit the most severe metabolic effects in mice. Thus, it looks like the five-generational WD-fed males have evolved to develop a protective mechanism in glucose and liver fat metabolism that can be inherited by the offspring.

Overall, our findings are in agreement with those of recent studies of multigenerational exposure performed in several animal models. For instance, in guppies, a wide range of plastic responses under different light conditions were observed, which were dependent on multigenerational exposure to different light environments (*Kranz et al., 2018*). In mites, zinc element sensitivity increased by continuous multigenerational exposure (*Jegede et al., 2019*). In mice, male sensitivity to environmental estrogens was enhanced by successive generations of exposure (*Horan et al., 2017*). Finally, rats undernourished for 50 generations showed multiple metabolic alterations that were not reversed in their respective F1 and F2 CD-fed progenies (*Hardikar et al., 2015*). Together, the present and previously published studies indicate that the exacerbation of stress-induced phenotypes upon multigeneration exposure as well as the stabilization of newly induced phenotypes is an evolutionarily conserved process. The underlying mechanism of this process is intriguing and worth to be explored in the future using this model since it might have not only social-medical implications, but also evolutionary perspectives.

Single-generation exposure to unhealthy diet strongly indicates that sperm RNAs are a possible epigenetic vector of intergenerational and transgenerational epigenetic inheritance of metabolic diseases (*Chen et al., 2016*; *Grandjean et al., 2016*; *Sharma et al., 2016*). However, these data do not exclude the possible involvement of epigenetic modifications, namely, DNA methylation, histone modifications, and chromatin structure alterations. This study takes a step further in this direction. Indeed, on the one hand, we showed that sncRNAs signature is not only the reflect of the diet of the father, as already demonstrated (*Nätt et al., 2019*; *Zhang et al., 2018*), but also that of the diet of the ancestors (*Figure 4*). Whether this is associated to spermatic epigenetic changes such as DNA methylation and chromatin structure alterations is an open question. On the other hand, our microinjection experiments showed that small RNAs are vectors of intergenerational inheritance but are not sufficient for the long-term inheritance of diet-induced metabolic alterations. In this context, our transcriptome profile of gWAT may provide important avenues to dissect the potential molecular mechanism(s) involved in this process, revealing an enrichment in genes potentially regulated by H3K4/K27 methylation and the PRC2 complex (*Figure 1—source data 2*).

Finally, in the present study, we focused our analyses on perigonadal adipose tissue, glucose/insulin sensitivity, and liver alterations. Considering the healthy and economic consequences of obesity and its comorbidities, such as cardiovascular diseases and fertility abnormalities, future studies will be important to determine the impact of multigenerational ancestor exposure on the development of obesity-associated comorbidities.

In conclusion, environmentally induced epigenetic modifications in germlines would contribute to the environmental adaptation and evolution of animal species. In the future, it will be important to assess how each epigenetic vector for inheritance interacts together to modulate the embryonic epigenome.

## Materials and methods

### Mice

All mouse experiments were performed with C57BL/6J mice obtained from Charles River (Charles River Laboratories, France). All mice were housed in a temperature-controlled system and maintained on a 12 hr light/dark cycle (lights on at 7 a.m.). Experimental mice were given ad libitum access to either a high-fat high-sugar diet (WD) (U8954 version 205 HF 45% of energy from fat, SAFE, France) or a CD (SAFE A04, 5% of energy from fat, SAFE, France) and sterile water. To evaluate the impact of the diet of paternal ancestors on metabolic health, we developed two experimental models. On the one hand, WD feeding was maintained for five successive generations through the paternal line. Briefly, 3-week-old male mice were divided into two groups. Males from the first group were kept on CD, and the males of the other group were fed a WD for 3 months. This first generation of CD-fed and WD-fed males was named CD1 and WD1, respectively. At 4 months old, two independent males of WD1 group were then crossed with 7-week-old C57BL/6J female mice (CD-fed) obtained from Charles River (Charles River Laboratories, France) to generate WD2 offspring. One or two litters were obtained per male. The male progenies were kept and subjected to the same experimental procedure. Then, at least one male, selected randomly, fed a WD for 3 months from each family was mated with CD-fed females to generate WD3 offspring. The same procedure was repeated twice to generate the WD4 and WD5 offspring (*Figure 1—figure supplement 1A*). However, from the WD2 generation, a considerable heterogeneity with respect to total body weight mass was observed within the same litter (*Figure 1—figure supplement 1C*). For this reason and as illustrated in *Figure 1—figure supplement 1A* starting from this generation, more than one male per litter was chosen to mate. The complete experimental design was performed twice at approximately 6 months' interval giving rise to four independent families. To demonstrate that we did not create significance with increasing the correlated sample, we selected a subset of 18 mice in WD4 and WD5 making sure that the four WD1 ancestors were equitably represented across the groups. Three combinations of 18 mice were selected and gave the same significant level with a p-value < 0.001 when total body weight mass or gWAT weight relative to total body weight of WD4 and WD5 groups was compared to the corresponding weight of CD group. One of these combinations is shown in *Figure 1—figure supplement 2B*. The same procedure was performed for the CD group (*Figure 1—figure supplement 1B*). However, in accordance with previous studies (*Cropley et al., 2016*; *Massiera et al., 2010*; *Fullston et al., 2015*; *Zhou et al., 2018*), body mass and gWAT weights of the CD-fed male progenies were very homogeneous (*Figure 1—figure supplement 2D*) and were combined in the same group.

Then, to determine whether the WD1 and WD5 phenotypes were paternally inter- and trans-generationally inherited in the absence of WD, four WD1 and nine WD5 males were crossed with a CD-fed female and their respective male and female progenies were fed a CD. The first generation was called F1-WD1 and F1-WD5, respectively. The F1 4-month-old male progenies were crossed with 7-week-old C57BL/6J female mice (CD-fed) to obtain the F2-WD1 and F2-WD5 progenies (*Figure 3A*). This experimental design was repeated once to obtain the F3-WD1 and F3-WD5 progenies. The control group of this experiment was obtained by crossing CD-fed males with 7-week-old C57BL/6J female mice (CD-fed) and maintained for four generations on CD.

To evaluate the role of sperm RNAs in transgenerational epigenetic inheritance of metabolic alterations, sperm RNAs extracted from two different CD, WD1, and WD5 males were microinjected into zygotes at the Center for Transgenic Models (University of Basel, Switzerland) following the

same procedure as described in *Gapp et al., 2014*. The resulting progenies were called F1-RNA-CD, F1-RNA-WD1, and F1-RNA-WD5 progenies, respectively. F2-RNA and F3-RNA progenies were obtained after crossing F1-RNA and F2-RNA 4-month-old males, respectively, with 7-week-old C57BL/6J female mice (CD-fed) obtained from Charles River (*Figure 5A*).

All mouse experiments were conducted in accordance with the National and European legislations for the care and use of research animals.

## Body weight and food intake

Body weights were measured every week from weaning until 5 months of age. Daily food consumption was estimated by weighing the remaining food every week.

For organ measurement, 5-month-old mice were anesthetized with sodium pentobarbital and rapidly dissected. Then, gonadal WAT, inguinal subcutaneous WAT, epididymis, liver, and kidneys were carefully isolated, cleaned of unrelated materials, and weighed. One part was fixed in 4% paraforamaldehyde (PFA), whereas the other one was snap frozen in liquid nitrogen.

## Blood metabolic parameter measurements

Blood metabolic parameters were detected under different physiological conditions, that is, a random-fed state and a 16 hr fasted state. Whole-blood glucose levels were determined using the One-Touch Vita (LifeScan, Johnson & Johnson company) system from tail blood. For plasma preparation, the blood was collected from the orbital sinus into sterile 1.5 ml tubes containing two drops of citrate sodium (3 M) and mixed gently. Blood cells were removed by centrifugation at $2000 \times g$ for 10 min at 4°C, and the resulting supernatant was immediately aliquoted and stored at −80°C. Serum CRP, leptin, adiponectin, and cholesterol levels were measured with the C-Reactive Protein ELISA (Mouse CRP, Elabscience, CliniSciences S.A.S., Nanterre, France), Leptin ELISA (ASSAYPRO, CliniSciences S.A.S., Nanterre, France), Adiponectin ELISA (mouse Adiponectin, EZMADP-60K, EMD Millipore Corporation, Darmsbalt, Germany), and Cholesterol Assay (Abcam, Paris, France) kits, respectively. All measurements were performed in accordance with the manufacturer's instructions.

## Glucose and insulin tolerance tests

Mice were placed in new cages prior to starvation. For GTTs, 12 hr fasted mice were injected intraperitoneal (i.p.) with a solution of sterile glucose (2 g/kg body weight) freshly prepared in 0.9% sterile saline. For ITTs, 6 hr fasted mice were injected i.p. with insulin diluted to 0.08 mU/μl in sterile saline for a final delivery of 0.8 mU/g body weight. Baseline glucose measurements were analyzed from tail blood before i.p. glucose or insulin injection (2 mg/g body weight) using the OneTouch Vita (LifeScan, Johnson & Johnson company) system. Blood glucose measurements were taken from the tail blood at the indicated points.

gWAT morphometry staining gWAT was fixed with Antigenfix (Microm Microtech, France), embedded in paraffin, sectioned, and stained with a hematoxylin and eosin (H&E) solution. Slides (4/group) were scanned with Axio-scan, which allowed the scanning of the entire slide at high resolution. Six pictures of six different areas from 1 to 2 sections per sample were chosen and analyzed with image analyzer software (ImageJ). Total areas of adipocytes were traced manually. The total count ranged from 3275 to 7052 adipocytes per condition. The mean surface area of the adipocytes was calculated using image analyzer software (ImageJ). For each sample, 400–1000 adipocytes were counted.

## Estimation of adipocyte number in gWAT

To estimate the number of adipocytes in gWAT depots, we applied a mathematical equation developed by Jo and colleagues (*Jo et al., 2009*), as previously described in *Gilleron et al., 2018*. Briefly, the number of adipocytes (N) was estimated by dividing the WAT mass (M) by the density of adipocytes (D = 915 g/l) multiplied by the mean volume of adipocytes within the WAT (V). The mean volume of adipocytes is calculated from the mean diameters of adipocytes, extracted from tissue sections images. The equation is presented below:

$$N = \frac{M}{\left(D \times \frac{4}{3} \times \pi r^3\right)}$$

## Computed tomography of mice

Anesthetized animals were placed in a SkyScan μCT-1178 X-ray tomograph (Bruker) and analyzed as previously described (*Beranger et al., 2014*). Mice were scanned using the following parameters: 104 μm pixel size, 49 kV, 0.5-mm-thick aluminum filter, and a rotation step of 0.9°. 3D reconstructions and analysis of whole abdominal fat were performed using NRecon and CTAn software (Skyscan), respectively, between thoracic 13 and sacral 4 vertebral markers.

## Liver TG measurement

Frozen small pieces of liver were placed in 2 ml tubes with Ceramic Beads (for Precellys homogenizer) and were homogenized in sodium acetate (0.2 M, pH 4.5) using the Precellys homogenizer. After centrifugation, the supernatant was stored at −80°C. The TGs in homogenates were measured according to the reagent kit instruction (Triglycerides FS, DiaSys Diagnostic Systems GmbH, Holzheim, Germany).

## Histological liver examination

Livers were prepared and fixed in 4% paraformaldehyde, embedded in paraffin, cut into 5-μm-thick slices, stained with H&E, mounted with neutral resins, and then scanned with Axio-scan, allowing the scanning of the entire slide at high resolution. Liver histology was blindly evaluated by two independent analyses using a semiquantitative scale adapted from previously validated procedures (*Nonalcoholic Steatohepatitis Clinical Research Network et al., 2005*). To that end, images from three different fields in each section were collected at ×20 magnification, and numbers of normal hepatocytes, microvesicular, and macrovesicular steatosis and degenerating hepatocytes were assessed.

## Sperm collection

Sperm were collected from the epididymis by squeezing. The cell suspension was centrifuged at 1000 rpm for 5 min, and the supernatant containing the spermatozoa was centrifuged at 3000 rpm for 15 min. To reduce contamination of somatic cells, the pellet was submitted to hypotonic shock by resuspension in water (250 μl), followed by the addition of 15 ml of PBS. The suspension was finally centrifuged at 3000 rpm for 15 min.

## Quantitative RT-PCR

Total RNA from epididymal adipose tissues was extracted using TRIzol reagent (Life Technologies, France) according to the manufacturer's instructions. Total RNA (0.5 μg) was reverse transcribed with mouse myeloblastosis virus reverse transcriptase (Promega) under standard conditions using hexanucleotide random primers according to the manufacturer's instructions. cDNA was amplified by PCR with specific primers. Real-time PCR was performed on the Light Cycler Instrument (Roche Diagnostics) using the Platinum SYBR Green kit (Invitrogen). Specific primers for mouse leptin and two mouse housekeeping genes used for normalization (*β-actin* and *34B4* mouse genes) were purchased from Sigma (Sigma, France). We used primers for *Leptin (forward,* AAC CTG GAA ATG CTC TGG CTGT; *reverse,* ACT CGC TGT GAA TGG CCT GAA A), *36B4F (forward,* TCC AGG CTT TGG GCA TCA; *reverse,* CTT TAT CAG CTG CAC ATC ACT CAG A), and *β-actin* (forward, CTA AGG CCA ACC GTG AAA AG; reverse, CCT GCT TCA CCA CCT TCT TG).

## RNA preparation and microinjection

Frozen sperm were stored at −80°C. RNA was then extracted by the TRIzol procedure (Invitrogen). The same preparations of sperm RNAs were used for microinjection and small RNA sequencing. RNA preparations were verified by spectrometry on an Agilent Bioanalyzer 2100 apparatus. Microinjection into fertilized eggs was performed as previously described in *Sarker et al., 2019*. RNA solutions were adjusted to a concentration of 1–2 μg/ml, and 1–2 pl were microinjected into the pronucleus of C57BL/6 fertilized mouse oocytes.

## Library preparation and sequencing

Total RNA was isolated from gonadal adipose tissue (eWAT; n = 9) samples using the Ambion Ribo-Pure (Thermo Fisher Scientific). RNA was quantified in a Nanodrop ND-1000 spectrophotometer, and RNA purity and integrity was checked by using a Bioanalyzer-2100 equipment (Agilent Technologies, Inc, Santa Clara, CA). Libraries were prepared using the TruSeq RNA Sample Preparation Kit (Illumina Inc, CA) and were paired-end sequenced (2 × 75 bp), by using the TruSeq SBS Kit v3-HS (Illumina Inc), in a HiSeq 2000 platform (Illumina Inc). More than 30 M PE reads were obtained for all samples.

## Transcriptomics analysis (RNA-sequencing analysis)

Raw sequence files were subjected to quality control analysis using FastQC. In order to avoid low-quality data, adapters were removed by Cutadapt and lower quality bases were trimmed by trimmomatic (*Bolger et al., 2014*). The quality-checked reads processed were mapped to the mouse reference genome GRCm38/mm10 using STAR (*Dobin et al., 2013*). Read abundance was evaluated for each gene followed by annotation versus mouse GTF by using the featureCounts function. The R package Edger was used in order to normalize the reads and identify DEGs (*McCarthy et al., 2012*). Genes with false discovery rate (FDR) < 0.05 after correcting for multiple testing were classified as DE (*Love et al., 2014*). The pheatmap and VolcanoPlot functions (R packages) were generated to graphically represent the expression levels (log2FC) and significance of DE genes among treatments. Next-generation sequence data have been deposited in the GEO Database with accession number (GSE148972) and a review access token (ovwzywcgnpublor).

## Small RNA-sequencing analysis

The experiment was carried out in triplicate. RNA libraries were prepared starting from 50 to 100 ng of total RNA from individual mice (n = 3 per group, three groups in total) and constructed using the Illumina TruSeq Stranded Small RNA Sequencing kit (Illumina Inc) according to the manufacturer's instructions. Sequencing was performed at the Genomix platform (Sophia-Antipolis, France) using the HiSeq 2500 (Illumina Inc).

Read quality was assessed using FastQC and trimmed, against known common Illumina adapter/primer sequences, using trimmomatic. The SmallRNAs UCAGenomix pipeline with Illumina adaptor trimming was used, read sizes < 15 nucleotides were discarded. In order to describe the general distribution of sperm sncRNAs, trimmed reads kept were mapped to small RNA database using a recently developed annotation pipeline, *SPORTS1.0* (*Shi et al., 2018*). We used the default settings and database files for the mouse genome GRCm38/mm10 that are available on the Sports github (https://github.com/junchaoshi/sports1.0). Averages summarized over biotypes were based on the default annotation result output files. Normalization of read abundance and differential expression analysis was performed by using DESeq R package on the Sports output files. The baseMean for each gene, the maximum of mean counts among all conditions, was at least 50 counts. Next generation sequencing (NGS) experiments have been deposited in the GEO Database with accession number (GSE138989).

## Statistics and reproducibility

Statistical analyses were performed using the Kruskal–Wallis test followed by the two-stage step-up method of Benjamini, Krieger, and Yekuteil for multiple comparisons of body weight, body composition, cholesterol, adiponectin, CRP, and leptin levels, as well as leptin mRNA expression and AUC-GTT and AUC-ITT within the WD cohorts, F1-, F2-, and F3-progenies and RNA-microinjected progenies. For each parameter, all groups were compared to each other in a single Kruskal–Wallis test followed by the two-stage step-up method of Benjamini, Krieger, and Yekuteil to adjust for all the performed multiple comparisons.

To determine whether the distribution of the total body weight mass was bimodal, we used the Hartigan's Dip Test for unimodality/multimodality available in the R Package 'diptest'. In this dip test, if p<0.05, we rejected the null hypothesis of unimodality and concluded that the distribution has more than one mode. Unless indicated otherwise, the unimodal distribution of total body weight mass was confirmed.

To measure the linear relationship between two variables, we used Spearman's correlation coefficient. All statistical analyses were performed with Prism 7 for Mac OS X software (GraphPad software, Inc). Data are presented as the median (IQR). $p_{adj} < 0.05$ was considered statistically significant.

Sample size and replicates are indicated in the figure legends. The WD cohort and WD progenies were repeated twice.

## Acknowledgements

We are grateful to Dr Jean-Jacques Remy for his careful help from the start of this project. We thank Drs Mireille Cormont, Sofia Fazio, Maria Stathopoulou, and Claire Mauduit for constructive discussions. We thank Marion Dussot for her technical assistance in performing the liver biochemistry. We relied on sequencing data generated by the IPMC Functional Genomics Facility (UCAGenomiX – IPMC platform; Sophia-Antipolis, France). We thank the Center for Transgenic Models (University of Basel, Switzerland) for the mouse microinjection assays. We are grateful to the C3M mouse facility (U1065, Nice). This work has been supported by ANR (grant# ANR-12-ADAPT-0022) and the FFAS 'Fonds Français pour l'Alimentation et la Santé' (15D52) and was partly supported by research funding from the Canceropôle PACA, Institut National du Cancer and Région Sud. FS was supported by the UCA-IDEX.

## Additional information

### Funding

| Funder | Grant reference number | Author |
| --- | --- | --- |
| Agence Nationale de la Recherche | NR-12-ADAPT-0022 | Georges Raad |
| Fonds Francais pour l'Alimentation et la Sante | 15D52 | Marie-Alix Derieppe |
| UCA-IDEX | | Fabrizio Serra |

The funders had no role in study design, data collection and interpretation, or the decision to submit the work for publication.

### Author contributions

Georges Raad, Valerie Grandjean, Conceptualization, Formal analysis, Supervision, Funding acquisition, Validation, Investigation, Visualization, Methodology, Writing - original draft, Writing - review and editing; Fabrizio Serra, Luc Martin, Michele Trabucchi, Formal analysis, Writing - review and editing; Marie-Alix Derieppe, Investigation, Methodology, Writing - review and editing; Jérôme Gilleron, Validation, Investigation, Methodology, Writing - review and editing; Vera L Costa, Validation, Methodology, Writing - review and editing; Didier F Pisani, Methodology, Writing - review and editing; Ez-Zoubir Amri, Formal analysis, Methodology, Writing - review and editing

### Author ORCIDs

Georges Raad https://orcid.org/0000-0001-8800-2796
Luc Martin https://orcid.org/0000-0001-5725-3955
Didier F Pisani https://orcid.org/0000-0001-5879-8527
Ez-Zoubir Amri https://orcid.org/0000-0001-8426-5396
Michele Trabucchi https://orcid.org/0000-0001-6885-5628
Valerie Grandjean https://orcid.org/0000-0003-1661-7411

### Ethics

Animal experimentation: All mouse experiments were conducted in accordance with the French and European legislations for the care and use of research animals. All of the animals were handled according to approved institutional animal care and use committee (APAFIS#8729-2017012716401597 (V7)) protocols (#381) of the Ministère de l'Enseignement Supérieur de la

Recherche et de l'innovation. The protocol was approved by the Committee on the Ethics of Animal Experiments of the University of Nice (Permit Number: 217-36).

## Decision letter and Author response

Decision letter https://doi.org/10.7554/eLife.61736.sa1
Author response https://doi.org/10.7554/eLife.61736.sa2

## Additional files

### Supplementary files

• Transparent reporting form

### Data availability

Sequencing data have been deposited in GEO under accession codes GSE138989 and GSE148972. All data generated or analyses during this study are included in the manuscript and supporting files.

The following datasets were generated:

| Author(s) | Year | Dataset title | Dataset URL | Database and Identifier |
|---|---|---|---|---|
| Serra F, Grandjean V | 2020 | Next Generation Sequencing Facilitates Quantitative Analysis of epididymal adipose tissue transcriptomes from mice fed either a control-diet or a Western-diet for one or five successive generations | https://www.ncbi.nlm.nih.gov/geo/query/acc.cgi?acc=GSE148972 | NCBI Gene Expression Omnibus, GSE148972 |
| Serra F, Grandjean V | 2019 | Small RNA-seq comparing transcriptome (small RNAs) of spermatozoa from mice fed either a control-Diet or a Western-Diet for One or Five successive generations | https://www.ncbi.nlm.nih.gov/geo/query/acc.cgi?acc=GSE138989 | NCBI Gene Expression Omnibus, GSE138989 |

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
