## [Decision Letter]

**Acceptance summary:**

The authors have explored the effect of five consecutive generations of a high-fat high-sugar diet in mice and their offspring's metabolic performance under a normal chow diet. It is very interesting that the chow-diet-fed progenies from these multigenerational western-diet-fed males develop a "healthy" overweight phenotype that persist for 4 subsequent generations. In parallel, the authors also performed zygotic sperm RNA injection using sperm RNAs from the western diet-fed males and showed that the sperm RNA indeed induce offspring metabolic phenotypes in F1 mice and some phenotypes persist to F2-F3, but none persist to F4, which is different from the mating induced phenotype. This study represents an advance for the mammalian epigenetic inheritance field.

**Decision letter after peer review:**

Thank you for submitting your article "Paternal multigenerational exposure to an obesogenic diet drives epigenetic predisposition to metabolic diseases in mice" for consideration by *eLife*. Your article has been reviewed by David James as the Senior Editor, a Reviewing Editor, and three reviewers. The reviewers have opted to remain anonymous.

The reviewers have discussed the reviews with one another and the Reviewing Editor has drafted this decision to help you prepare a revised submission.

Summary:

In this manuscript, Raad and colleagues exposed male mice to a western diet before conception for 5 consecutive generations and measured body weight, adiposity and various metabolic markers in the offspring. Sequencing of small RNA in sperm from founders identified several differentially expressed tRF and miRNA species. Microinjection of RNAs recapitulated some, but not all effects on body weight and metabolism. The authors report an aggravation of adiposity along generations and a phenotype that persists for 4 consequent generations. Such persistence of phenotype was not observed in animals originating from microinjection of total RNAs, suggesting other epigenetic mechanisms are at play in the persistence of phenotype. Overall the studies were considered to be of interest by the referees but one major overarching problem identified by them concerned the study design and the statistical analyses that limited interpretation of the study. These issues need to be seriously addressed by the authors in a major revision. These and other points are listed below.

Essential revisions:

1. In line 72, the authors state that "the average body weight of the WD-fed male mice increased gradually with multigenerational WD feeding", however, the results of the test indicating gradual increase is not reported. As described in the legend of Figure 1, the test performed tested differences in body weight between the control group and each individual generation, not the generations to each other. Visually, it rather seems that in fact, body weight was not gradually increased for instance, comparison of WD1 and WD3, or WD2 and WD5, does not support the "gradual increase" in body weight that the authors are claiming.

2. There is a lack of clarity in the Materials and methods in regards to numbers of animals used in each generation, the number of founders, and what constitutes the control group. In the legend of Figure 1, it is stated that 5 males were used from WD2 and on. However, the method section states "(…) 4 to 6 independent males of WD1 group". The reviewer assumes that the authors know how many animals were used in the WD1 group, and that the authors meant 4 to 6 animals per WD generation. However, if the details indicated in the legend of Figure 1 are accurate (5 fathers per group from WD2), how is it possible that 4 to 6 animals were used? The reviewer suggests to clarify this in the text, as well as in a more detailed experimental setup diagram/schematic stating the number of fathers in each generation, the number of offspring studied in each litter, and the total number of offspring studied for each generation.

3. In Supplemental Figure 1I, the CD1 group appears to be composed of 7 individuals and the CD2 group of 10 individuals. This is not consistent with the numbers reported in Figure 1A (10 in CD1 and 13 in WD3) and Figure 1B (22 visible dots). It is thus difficult for the reviewer to trust that body weights were truly compared between all animals in CD1 and CD5. Regardless, the reviewer is intrigued by the choice of the authors to only study control animals from the first generation (CD1), and the fifth generation (CD5) offspring, as they describe in the methods that, for the control group, they followed the same procedure as the WD group, which should have led to the generation of control animals in all F1, F2, F3, F4 and F5 generations. The authors should clarify on this, and if they indeed generated these animals, they should use body weight data in each generation of controls and compare them to their respective generation WD group (i.e. CD1 to WD1, CD2 to WD2 etc..). By having different sample size in the various groups, the authors are biasing results of the statistical test being made, as greater sample size is likely to compare statistically different than a group with lower sample size (as with CD(22 observations) and WD2(12observations) in Figure 1B, but also with the RNA-seq results). In the same line, there were more animals studied in WD4 and WD5 compared to WD1-3 which is likely biasing statistical analysis. Again, if the study design described in the methods section is accurately reported, it implies that an average of 3 offspring per fathers were used in WD1-3, and 8-10 (a full litter) for the WD4-5.

4. Number of mice per group range widely, and it is unclear how many matings this represents. Figure 3 legend states 4 WD1 and 9 WD5 males from different littermates were mated with CD females – again, unclear – do you mean from different litters? Numbers shown in panel A do not seem to concur with those in panels B, C.

5. It is unclear why mice were studied at the various ages- eg Across data sets, ages shown range from 10 weeks, 12 weeks, 16 weeks, 18 weeks. Note there are inconsistencies regarding figure formats and some details are missing, which makes it hard to understand what the authors found. Figure S3 and S5- no n values given. Labels in S4 D, E hard to follow.

6. In several of the figures, it is not clear what the significance (*) is being compared to – is it always CD? Eg Figure 3, Figure 4

7. It appears that variability increases from WD1 to WD5- with larger ranges evident- is this why n increases across generations? And is this a consistent observation across paternal studies of this kind?

8. Regarding the phenotype induced by sperm RNA injection, the description should be more precise as the current description is not all consistent with the data presented. In Figure.4, some parameter changes persist to F2-F3, this already suggest transgenerational inheritance rather than merely intergenerational transmission. The more precise description should be that sperm RNAs can unequivocally induce intergenerational phenotype, but may induce some transgenerational features – although the effect is weaker than the effect induced by whole sperm. In fact, in a previous study using a mental-stress induced model, sperm RNA injection can also induce phenotype in both F1 and F2 generations (Nat Neurosci. 2014 May;17(5):667-9.).

9. The sperm small RNA analysis part (Figure S4) is relatively weak. The datasets generated are in fact quite valuable as they include the sperm from control diet, first-generation WD and the Fifth-generation WD. This is an opportunity to explore the difference especially between the first-generation WD and Fifth-generation WD as no one has done this before. The current data analyses are crude and did not show these differences in an informative way. It is needed to at least provide the overall length distribution of each datasets with the annotation of different types of small RNAs. The authors have shown some difference regarding miRNAs and tRNA-derived small RNAs (tsRNAs) in Figure S4, it would be interesting to also look at the rRNA-derived small RNAs (rsRNAs) because rsRNAs are also extensively discovered in both mouse and human sperm and these sperm rsRNAs are sensitive to dietary changes (Nat Cell Biol. 2018 May;20(5):535-540; PLoS Biol. 2019 Dec 26;17(12):e3000559.), closely associated with mammalian epigenetic inheritance and thus represent a component of the recently proposed sperm RNA code in epigenetic inheritance (Nat Rev Endocrinol. 2019 Aug;15(8):489-498). The reanalysis of the datasets could be done by SPORTS1.0 (Genomics Proteomics Bioinformatics. 2018 Apr;16(2):144-151.), which provide the annotation and analyses of miRNAs, tsRNA, rsRNAs and piRNAs that have been used in the above mentioned publications (Nat Cell Biol. 2018 May;20(5):535-540; PLoS Biol. 2019 Dec 26;17(12):e3000559)

Revisions expected in follow-up work:

A much more detailed and thorough description of the experimental design with the possible inclusion of a schematic.

A better explanation of statistical analysis with a possible reanalysis of existing data.

Reanalysis of data according to Point #9 above.

[Editors' note: further revisions were suggested prior to acceptance, as described below.]

Thank you for submitting your article "Paternal multigenerational exposure to an obesogenic diet drives epigenetic predisposition to metabolic diseases in mice" for consideration by *eLife*. Your article has been reviewed by David James as the Senior Editor, a Reviewing Editor, and three reviewers. The reviewers have opted to remain anonymous.

The reviewers have discussed the reviews with one another and the Reviewing Editor has drafted this decision to help you prepare a revised submission.

All reviewers felt that the manuscript was somewhat improved but significant problems still remain that need to be addressed. In particular several referees mentioned that statistical analyses and their presentation remain a problem, and this needs to be rectified before we can move forward. Because of this we sought the opinion of a fourth referee who is an expert in Biostatistics, and their report is attached below. This individual raised similar concerns but felt that with appropriate revisions these problems might be overcome. This referee also provides comments for your help on some of the original Reviewer's feedback. You may in fact find it useful to consult with a Biostatistician to help address some of these problems. Please give these issues serious consideration before resubmitting your revised manuscript.

Reviewer #4

The current manuscript studied the consequences of multigenerational (5 consecutive generations) of unbalanced diet feeding (WD) on the metabolic health of four subsequent generations of chow-diet-fed offsprings.

My review focuses primarily on the validity of the statistical analysis presented in this paper. In general, the analytics is appropriate but the presentation is confusing. Much of the statistical methods are written inside the caption and it's not very clear. Most of my comments below are related to improving the readability of the manuscript.

In summary, transcriptomics analysis (RNA-sequencing analysis) pipeline is standard and appropriate. Small RNA-sequencing analysis and annotation are based on SPORTS1.0 with DEseq used for DE analysis. Majority of statistical analyses were performed using the Kruskal-Wallis test followed by the two-stage step-up method of Benjamini, Krieger and Yekuteil for multiple comparisons using GraphPad seems OK, but given a large number of comparisons, the multiple adjustments are really within WD1 to WD5 correct? Given the small n value in a certain context, caution is needed regarding the p-value interpretation.

Results presentations can be improved

- It is not clear what the significance (*) is being compared to – is it always CD? The caption wrote, "and different numbers (1,2, 3) label significance when comparing the WDs groups with CD and with WD1, WD2, WD3, respectively." For example in Figure 1B, significance 1, 3 for WD4, is it compared with CD or WD1? Regardless, there looks like a bimodal distribution (a mixture of two groups) in WD4 in Figure 1B and the significant differences is driven by the sub-populations?

- Figure 3E demonstrate a clear bimodal distribution for the F3 cohort, and one should be extra careful on the p-value interpretation. These comments apply to a few other sub-figures.

- For Table 1, not sure why some of the results were mean+/- SD and others were median? Was the multiple comparison-adjusted across a mixture of median and mean or just one centralization measure.

Reviewer #1 comment 1

Authors address this issue – the term "gradually" was removed. There is a difference in the later generation but probably not before.

Reviewer #1 comment 2

Page 38 (Figure 1—figure supplement 1) has the full exp design, however, the selection process is random (stated in the material and methods).

Reviewer #1 comment 3 – two key issues here:

a. All n = 53 (in WD5) is not independent.

Note, the design difference between WD3 and WD4-5, is that prior to WD3, only one mice from each litter is selected to mate whereas going from WD3 to WD4, there is no selection, nearly all litters (14 out of the 17 litter from WD3) were chosen to mate.

b. Why only CD1 an CD5 and not CD1, CD2,.… CD5. Should WD1 be compared to corresponding CD 1 rather than mixing CDs all together.

Re point (a) Reviewer #1 is concerned over the variability coming from (n=4) parents in WD1 is different from the variability coming from WD5. If you look at Figure 1—figure supplement 1A, the argument is mice from the same ancestor is likely to be more similar? The n isn't really as large as the authors have argued. One way to deal with this doing "repeated measurement ANOVA" accounting for the fact that mice from the same ancestor (same WD1) is dependent. Another adhoc way to demonstrate you didn't create significance with increasing the correlated sample is to select a subset in WD4 (e.g. n4 = 17) and WD5 to perform the analysis.

Re point b I think as far as body weights go, CDs (23 dots) looks very tight in Figure 1B, splitting them probably won't make a big difference.

[Editors' note: further revisions were suggested prior to acceptance, as described below.]

Thank you for resubmitting your work entitled "Paternal multigenerational exposure to an obesogenic diet drives epigenetic predisposition to metabolic diseases in mice" for further consideration by *eLife*. Your revised article has been evaluated by David James (Senior Editor) and a Reviewing Editor.

As before, the only issues concern the statistical analysis and so I opted to go back again to the original stats expert reviewer who also sought advice from an additional stats person. They considered the manuscript to be much improved but there are some relatively minor remaining issues that need to be addressed, as outlined below.

It is my opinion that these can be dealt with by you in a relatively straightforward manner but this will/may require some data re-analysis. I recommend that you consult with a statistician to help in addressing these comments if you are uncertain what the referee is asking.

Reviewer #4:

The manuscript is much improved, but I have one comment.

My previous comment on data heterogeneity did not intend to seek further statistical tests but for the authors to look deeper into their data. In particular, if there exists a subpopulation, how does that impact or change your conclusion. Furthermore, with the small sample size, it is difficult to conclude deviation from unimodal modality or outliers' existence.

In the author's response. I disagree that using a permutation test or a non-parametric statistical test justify investigating the mean in a bimodal distribution. The author's reply misunderstood my focus. There are two concepts here.

The appropriateness of a chosen statistics (eg difference in means), which is dependent on your question of interest and interpretation; and

The appropriateness of the distribution of statistics, which guides the p-value conclusion.

The use of non-parametric test deals with the latter but doesn't deal with the former. I question the interpretation of "comparing two different means" when the data is not unimodal. Here, there is a difference between what is statistically correct versus what is a reasonable assumption in practice. For data with a bimodal distribution, if you believe that there is only one population with a bimodal distribution, then non-parametric testing will provide the required solutions to compare the means. However, in practice, a more reasonable assumption when we observe bimodality or large spread in data (on the boxplot) is that this reflects two sub-populations suggesting latent classes. Here, care needs to be taken in the interpretation of your results. For example, if the data from group A has two modes (two sub-population) with two means of 4 and 8, respectively, and the data from group B has a mean of 4. Concluding that group B has a high overall average than group A doesn't capture the underlying latent structure. It isn't appropriate for the subpopulation in group A with a mean of 4.

Looking at Figure 1B and Figure 3, the data shows a huge spread, and this heterogeneity is either due to subpopulations or possibly outliers that you need to account for. Rather than a whole series of Dips test, some careful examination of the small number of cases where the data looks potentially bimodal or highly variable and describe how your overall interpretation would be alternated would be more informative than trying simply to justify interpretation based on the overall mean. I suggest performing sensitivity analysis and/or adding such a caveat in the discussion of your results and conclusion.

---

## [Author Response]

Essential revisions:1. In line 72, the authors state that "the average body weight of the WD-fed male mice increased gradually with multigenerational WD feeding", however, the results of the test indicating gradual increase is not reported. As described in the legend of Figure 1, the test performed tested differences in body weight between the control group and each individual generation, not the generations to each other. Visually, it rather seems that in fact, body weight was not gradually increased for instance, comparison of WD1 and WD3, or WD2 and WD5, does not support the "gradual increase" in body weight that the authors are claiming.

We agree with the reviewer that the term “gradually” was not appropriate therefore was removed. What our study really shows is an exacerbation of the overweight phenotype upon successive WD-feeding generations. This exacerbation is evidenced by the fact that the males of WD4 and WD5 groups weighted significantly more than those of the WD1 group. This clarification has been included in the revised version lines 68-72 and stated in the legend of the Figure 1. This conclusion is supported by comparing differences in body weight among the different generations to each other (Figure 1B), thanks to a Kruskal-Wallis test, a rank-based nonparametric test, followed by the two-stage linear step-up procedure of Benjamin, Krieger and Yekutieli for multiple comparisons.

2. There is a lack of clarity in the methods in regards to numbers of animals used in each generation, the number of founders, and what constitutes the control group. In the legend of Figure 1, it is stated that 5 males were used from WD2 and on. However, the method section states "(…) 4 to 6 independent males of WD1 group". The reviewer assumes that the authors know how many animals were used in the WD1 group, and that the authors meant 4 to 6 animals per WD generation. However, if the details indicated in the legend of Figure 1 are accurate (5 fathers per group from WD2), how is it possible that 4 to 6 animals were used? The reviewer suggests to clarify this in the text, as well as in a more detailed experimental setup diagram/schematic stating the number of fathers in each generation, the number of offspring studied in each litter, and the total number of offspring studied for each generation.

To clarify our procedure, we added a detailed experimental setup (Figure1—figure supplement 1) and changed the text in both the legend of Figure 1 and in Materials and methods (Lines 326-336).

3. In Supplemental Figure 1I, the CD1 group appears to be composed of 7 individuals and the CD2 group of 10 individuals. This is not consistent with the numbers reported in Figure 1A (10 in CD1 and 13 in WD3) and Figure 1B (22 visible dots). It is thus difficult for the reviewer to trust that body weights were truly compared between all animals in CD1 and CD5.

We apologize for that. The number of CD-fed mice obtained at the F1 and F5 generations were, as indicated in Figure 1A, 10 and 13, respectively. The Figure 1B has now been corrected.

Regardless, the reviewer is intrigued by the choice of the authors to only study control animals from the first generation (CD1), and the fifth generation (CD5) offspring, as they describe in the methods that, for the control group, they followed the same procedure as the WD group, which should have led to the generation of control animals in all F1, F2, F3, F4 and F5 generations. The authors should clarify on this, and if they indeed generated these animals, they should use body weight data in each generation of controls and compare them to their respective generation WD group (i.e. CD1 to WD1, CD2 to WD2 etc..).

We thank the reviewer for raising this important point. In fact, to evaluate the evolution of body weight mass over successive generations of WD feeding, we used only one CD group as was the case in multigenerational studies (1, 2).

This strategy is based on previous studies showing that CD-fed generations exhibit very similar metabolic profiles such as total body weight mass and other metabolic parameters, ie fat mass, leptin and glucose sensitivity (3, 4).

Before mixing the control groups, body weight mass among all CD-fed generations were analyzed. No statistical difference (p>0.05) was observed when we applied a Kruskal-Wallis test, a rank-based nonparametric test, two-stage linear step-up procedure of Benjamin, Krieger and Yekutieli for multiple comparisons (Figure 1—figure supplement 2J).

Finally, because only the first and fifth generation offspring were used to analyze organ weights (Figure 1B), we used a control offspring group of these two generations.

By having different sample size in the various groups, the authors are biasing results of the statistical test being made, as greater sample size is likely to compare statistically different than a group with lower sample size (as with CD(22 observations) and WD2(12observations) in Figure 1B, but also with the RNA-seq results). In the same line, there were more animals studied in WD4 and WD5 compared to WD1-3 which is likely biasing statistical analysis.

We agree with the reviewer that increasing our sample size can give us greater power to detect differences. Accordingly, in our analysis, we took into account unequal size of the sample by using a Kruskal-Wallis test followed by two-stage linear step-up procedure of Benjamin, Krieger and Yekutieli which can be used for multiple comparisons of independent sample of unequal size (5).

Again, if the study design described in the Materials and methods is accurately reported, it implies that an average of 3 offspring per fathers were used in WD1-3, and 8-10 (a full litter) for the WD4-5.

Indeed, the study design described in the methods section was inaccurately reported and has been corrected in the new version lines 326-336 and in Figure 1A—figure supplement 1. Because of the heterogeneity of the progenies, we, in fact, used 1, 2, 3 progenitors per offspring in WD1-5 generations. That explains why there were more animals studied in WD4 and WD5 compared to WD1-3.

4. Number of mice per group range widely, and it is unclear how many matings this represents. Figure 3 legend states 4 WD1 and 9 WD5 males from different littermates were mated with CD females – again, unclear – do you mean from different litters?

Yes, sorry we meant from different litters. The 4 WD1 males correspond to the male used in the Figure 1. Also, the 9 WD5 male were chosen arbitrary from the 4 different families. In the revised manuscript, we have clarified this point in the Figure 3 legend.

Numbers shown in panel A do not seem to concur with those in panels B, C.

In the revised manuscript, we corrected this mistake.

5. It is unclear why mice were studied at the various ages- eg Across data sets, ages shown range from 10 weeks, 12 weeks, 16 weeks, 18 weeks.

We apologize about that. No reason justified the use of various ages. The data have been homogenized. In all figures and Figures-Source data, body weight mass was indicated at 12 and 16 weeks. The other parameters such as organ weight and abdominal adipose volume have been measured at the time of sacrifice of the mice, namely 18 weeks. This was made explicit now in the legends of the source data.

Note there are inconsistencies regarding figure formats and some details are missing, which makes it hard to understand what the authors found. Figure S3 and S5- no n values given. Labels in S4 D, E hard to follow.

We have now changed the labels in Figure S3 (now Figure 3—figure supplement 4), and the police size in Figure S5 (now Figure 4). The n values are now indicated in the legend.

6. In several of the figures, it is not clear what the significance (*) is being compared to – is it always CD? Eg Figure 3, Figure 4

Yes, (*) always indicated the comparison between CD and experimental groups. We now clearly stated this in the Legend of Figure 3 and Figure 5.

7. It appears that variability increases from WD1 to WD5- with larger ranges evident- is this why n increases across generations?

We fully agree with the reviewer that the variability/heterogeneity increases from WD1 to WD5. This is the reason why n increases across generations. Indeed, since we observed heterogeneity among the same litter and to limit sampling bias and to keep the number of 4 independent families, we crossed more than one male from each litter to obtain the following generation. This precision has been added to the legend of Figure 1—figure supplement 1.

And is this a consistent observation across paternal studies of this kind?

In our knowledge, very few studies have been conducted on this condition in mice (2, 6, 7). In particular, the study of Horan et al. (2017)(6) suggests that maintaining for more than one generation the new environment tends to accelerate the variability of the population.

8. Regarding the phenotype induced by sperm RNA injection, the description should be more precise as the current description is not all consistent with the data presented. In Figure.4, some parameter changes persist to F2-F3, this already suggest transgenerational inheritance rather than merely intergenerational transmission. The more precise description should be that sperm RNAs can unequivocally induce intergenerational phenotype, but may induce some transgenerational features – although the effect is weaker than the effect induced by whole sperm. In fact, in a previous study using a mental-stress induced model, sperm RNA injection can also induce phenotype in both F1 and F2 generations (Nat Neurosci. 2014 May;17(5):667-9.).

We agree with the reviewer and changed the text accordingly lines 211-221 and 231-236.

9. The sperm small RNA analysis part (Figure S4) is relatively weak. The datasets generated are in fact quite valuable as they include the sperm from control diet, first-generation WD and the Fifth-generation WD. This is an opportunity to explore the difference especially between the first-generation WD and Fifth-generation WD as no one has done this before. The current data analyses are crude and did not show these differences in an informative way. It is needed to at least provide the overall length distribution of each datasets with the annotation of different types of small RNAs. The authors have shown some difference regarding miRNAs and tRNA-derived small RNAs (tsRNAs) in Figure S4, it would be interesting to also look at the rRNA-derived small RNAs (rsRNAs) because rsRNAs are also extensively discovered in both mouse and human sperm and these sperm rsRNAs are sensitive to dietary changes (Nat Cell Biol. 2018 May;20(5):535-540; PLoS Biol. 2019 Dec 26;17(12):e3000559.), closely associated with mammalian epigenetic inheritance and thus represent a component of the recently proposed sperm RNA code in epigenetic inheritance (Nat Rev Endocrinol. 2019 Aug;15(8):489-498). The reanalysis of the datasets could be done by SPORTS1.0 (Genomics Proteomics Bioinformatics. 2018 Apr;16(2):144-151.), which provide the annotation and analyses of miRNAs, tsRNA, rsRNAs and piRNAs that have been used in the above mentioned publications (Nat Cell Biol. 2018 May;20(5):535-540; PLoS Biol. 2019 Dec 26;17(12):e3000559)

We thank the reviewer for this very helpful comment. The analysis which is now included in our manuscript (lines 173-196) and Figure 4, might shed some light on the understanding of the mechanism of epigenetic inheritance showing that the sensitivity of sperm sncRNAs signature to diet observed independently by several groups (8, 9) is also modulated by the diet of the ancestors.

Revisions expected in follow-up work:A much more detailed and thorough description of the experimental design with the possible inclusion of a schematicA better explanation of statistical analysis with a possible reanalysis of existing dataReanalysis of data according to Point #9 above.

We would like to thank you for the valuable comments. They were immensely helpful in enhancing the quality of the manuscript. The 3 points were discussed in the present letter and the requested modifications were added to the manuscript.

[Editors' note: further revisions were suggested prior to acceptance, as described below.]

Reviewer #4The current manuscript studied the consequences of multigenerational (5 consecutive generations) of unbalanced diet feeding (WD) on the metabolic health of four subsequent generations of chow-diet-fed offsprings.My review focuses primarily on the validity of the statistical analysis presented in this paper. In general, the analytics is appropriate but the presentation is confusing. Much of the statistical methods are written inside the caption and it's not very clear. Most of my comments below are related to improving the readability of the manuscript.In summary, transcriptomics analysis (RNA-sequencing analysis) pipeline is standard and appropriate. Small RNA-sequencing analysis and annotation are based on SPORTS1.0 with DEseq used for DE analysis. Majority of statistical analyses were performed using the Kruskal-Wallis test followed by the two-stage step-up method of Benjamini, Krieger and Yekuteil for multiple comparisons using GraphPad seems OK, but given a large number of comparisons, the multiple adjustments are really within WD1 to WD5 correct? Given the small n value in a certain context, caution is needed regarding the p-value interpretation.

Yes, the multiple adjustments are within WD1 to WD5. Thus, CD and WDs groups were compared to each other using a Kruskal-Wallis test followed by the two-stage step-up method of Benjamini, Krieger and Yekuteil to adjust for all the multiple comparisons. This test was performed separately for each parameter. That is, now, clearly stated in the Material and methods, line 514-516.

Results presentations can be improved- It is not clear what the significance (*) is being compared to – is it always CD? The caption wrote, "and different numbers (1,2, 3) label significance when comparing the WDs groups with CD and with WD1, WD2, WD3, respectively." For example in Figure 1B, significance 1, 3 for WD4, is it compared with CD or WD1?

In Figure 1, Figure 3 and Figure 3—figure supplement1, groups that are significantly different are now clearly identified with line to make the figures easier to understand.

Regardless, there looks like a bimodal distribution (a mixture of two groups) in WD4 in Figure 1B and the significant differences is driven by the sub-populations?

Thanks for this valuable comment. As now indicated in the Materials and methods (lines 517-521), the unimodal distribution of body weight for WDs groups was tested using the Hartigans’ Dip Test for unimodality/ multimodality available in the R Package ‘diptest’, under the null hypothesis of unimodality. Since all p-values are above 0.1 (see below), we can make the hypothesis that all WD groups follow a unimodal distribution.

WD1: p-value = 0.1168

WD2: p-value = 0.2798

WD3: p-value = 0.8995

WD4: p-value = 0.9311

WD5: p-value = 0.6348

- Figure 3E demonstrate a clear bimodal distribution for the F3 cohort, and one should be extra careful on the p-value interpretation. These comments apply to a few other sub-figures.

Thanks for this valuable comment. Indeed, the distribution of the F2/F3-WD may be suggestive of a bimodal distribution. So as suggested by the reviewer, the unimodality / multimodality of distributions for body weight for all groups analyzed in Figure 3 was tested using the Hartigans’ Dip Test for unimodality/ multimodality. As shown below, only one group did not follow a unimodal distribution, namely the F1-WD1 group. This observation has now been added in the text of the Legend of the Figure 3. We have, nevertheless, maintained the analysis that we performed and the significant results as we have used a non-parametric statistical test, which allows us to make comparisons without any assumptions about the data distributions.

WD male progenies

F1-WD1: p-value = 0.03723

F2-WD1: p-value = 0.9543

F3-WD1_1: p-value = 0.08312

F4-WD1: p-value = 0.5875

F1-WD5: p-value = 0.8995

F2-WD5: p-value = 0.9282

F3-WD5: p-value = 0.7437

F4-WD5: p-value = 0.08134

WD female progenies

F1-WD1_F: p-value = 0.2048

F3-WD1_2: p-value = 0.7197

F2-WD1_F: p-value = 0.5985

F1-WD5_F: p-value = 0.5355

F2-WD5_F: p-value = 0.8516

F3-WD5_F: p-value = 0.754

F4-WD5F: p-value = 0.2112

- For Table 1, not sure why some of the results were mean+/- SD and others were median?

We apologize for this lack of clarity in the legend. As now specified, results are presented as median (IQR) in Table 1, Figure 1-source data 1, Figure 3—source data 3, 4 and Figure 5—source data 7-10.

Was the multiple comparison-adjusted across a mixture of median and mean or just one centralization measure.

The multiple comparison was adjusted across one centralization measure, namely the median.

Reviewer #1 comment 1Authors address this issue – the term "gradually" was removed. There is a difference in the later generation but probably not before.

This precision has been added in the revised manuscript, line 71 – 75. Here, we found that whereas multigenerational WD feeding had no measurable impact on total mass body weight on the first 3 generations (WD1 to WD3), the WD4 and WD5 males weighted significantly more than the WD1 and WD3 males (p<0.05 and p<0.01, respectively) (Figure 1B and Figure 1—figure supplement 2A, B).

Reviewer #1 comment 2Page 38 (Figure 1—figure supplement 1) has the full exp design, however, the selection process is random (stated in the material and methods).

The full experimental design has now been fully explained. In particular, our choice to select more than one male/ litter is now clearly stated in Material and methods (lines 344-347). We wrote that “from the WD2 generation, a considerable heterogeneity with respect to total body weight mass was observed within the same litter (Figure 1—figure supplement 1C). For this reason and as illustrated in Figure 1—figure supplement 1A, starting from this generation, more than one male per litter was chosen to mate.”

Reviewer #1 comment 3a. All n = 53 (in WD5) is not independent.Note, the design difference between WD3 and WD4-5, is that prior to WD3, only one mice from each litter is selected to mate whereas going from WD3 to WD4, there is no selection, nearly all litters (14 out of the 17 litter from WD3) were chosen to mate.

Yes, that is true that the WD4 and WD5 were not independent. To better explain this experiment procedure, we indeed added this observation in the legend of the figure 1, figure supplement 1 and explained the reason of this change which was the following: because of the heterogeneity of the total body weight mass within a litter it was difficult to make the choice of which male to mate.

b. Why only CD1 an CD5 and not CD1, CD2,.… CD5.

We mixed only CD1 and CD5 males because only these 2 generations of CD males were extensively analyzed (organ weights, GTT and ITT experiments).

Should WD1 be compared to corresponding CD 1 rather than mixing CDs all together.

As shown in Figure 1—figure supplement 2C, we compared the total body weight mass of each WD group to their respective control group (WD1 versus CD1, WD2 vs CD2….).

Re point (a) reviewer #1 is concerned over the variability coming from (n=4) parents in WD1 is different from the variability coming from WD5. If you look at Figure 1—figure supplement 1A, the argument is mice from the same ancestor is likely to be more similar? The n isn't really as large as the authors have argued. One way to deal with this doing "repeated measurement ANOVA" accounting for the fact that mice from the same ancestor (same WD1) is dependent. Another adhoc way to demonstrate you didn't create significance with increasing the correlated sample is to select a subset in WD4 (e.g. n4 = 17) and WD5 to perform the analysis.

We would like to thank the reviewer for this suggestion. We agree with both reviewers that genetically and epigenetically speaking mice from the same ancestor is likely to be more similar than mice from different ancestors. To demonstrate that the significance is real and not affected by increasing the correlated sample, we selected a subset of 18 mice in WD4 and WD5 making sure that the 4 WD1 ancestors were equitably represented across the groups. 3 combinations of 18 mice were selected. All of these combinations gave the same conclusions, namely, the total body weight mass of the first three generations was different from the control group but very similar within the WD groups. In addition, the fourth and fifth generations were significantly different from CD and WD1 groups. One of these combinations is now presented in Figure 1—figure supplement 2B.

We use this option since while the total body weight mass appeared heterogeneous within a litter, this parameter was quite homogeneous within the same WD generation as now showed in Figure 1—figure supplement 1C.

Re point (b) I think as far as body weights go, CDs (23 dots) looks very tight in Figure 1B, splitting them probably won't make a big difference.

Yes, we agree with the reviewer. Indeed, as shown in Figures 1—figure supplement 2C, the comparison between each CD group and their respective WD does not change the significance level between WD and CD groups.

[Editors' note: further revisions were suggested prior to acceptance, as described below.]

Reviewer #4:The manuscript is much improved, but I have one comment.My previous comment on data heterogeneity did not intend to seek further statistical tests but for the authors to look deeper into their data. In particular, if there exists a subpopulation, how does that impact or change your conclusion. Furthermore, with the small sample size, it is difficult to conclude deviation from unimodal modality or outliers' existence.In the author's response. I disagree that using a permutation test or a non-parametric statistical test justify investigating the mean in a bimodal distribution. The author's reply misunderstood my focus. There are two concepts here.The appropriateness of a chosen statistics (eg difference in means), which is dependent on your question of interest and interpretation; andThe appropriateness of the distribution of statistics, which guides the p-value conclusion.The use of non-parametric test deals with the latter but doesn't deal with the former. I question the interpretation of "comparing two different means" when the data is not unimodal. Here, there is a difference between what is statistically correct versus what is a reasonable assumption in practice. For data with a bimodal distribution, if you believe that there is only one population with a bimodal distribution, then non-parametric testing will provide the required solutions to compare the means. However, in practice, a more reasonable assumption when we observe bimodality or large spread in data (on the boxplot) is that this reflects two sub-populations suggesting latent classes. Here, care needs to be taken in the interpretation of your results. For example, if the data from group A has two modes (two sub-population) with two means of 4 and 8, respectively, and the data from group B has a mean of 4. Concluding that group B has a high overall average than group A doesn't capture the underlying latent structure. It isn't appropriate for the subpopulation in group A with a mean of 4.Looking at Figure 1B and Figure 3, the data shows a huge spread, and this heterogeneity is either due to subpopulations or possibly outliers that you need to account for. Rather than a whole series of Dips test, some careful examination of the small number of cases where the data looks potentially bimodal or highly variable and describe how your overall interpretation would be alternated would be more informative than trying simply to justify interpretation based on the overall mean. I suggest performing sensitivity analysis and/or adding such a caveat in the discussion of your results and conclusion.

Based on Reviewer’s suggestions, we have now added several sentences to highlight phenotypic heterogeneity in the context of our model of paternal multigenerational exposure to Western diet. Please find here below, the sentences and the references added in the manuscript:

Lines 71-78

Despite marked heterogeneity in the WD4 and WD5 populations, we found that the WD4 and WD5 males weighted significantly more than the WD1 and WD3 ones (p<0.05 and p<0.01, respectively) (Figure 1B and Figure 1—figure supplement 2A, B). Interestingly, growing heterogeneity of the body weight mass between the males of the first and the latter generations (Figure 1—figure supplement 1C) was observed in the 4 independent families, indicating that the phenotypic heterogeneity previously observed in diet-induced obesity mouse models (10) increases progressively over the generations.

Lines 168-172

Thus, as illustrated in Figure 3B-I, the populations of males and females of the F3-WD1 progenies were very homogeneous, exhibiting metabolic characteristics very similar to control mice. By contrast, both populations of males and females of the F3-WD5 progenies were heterogeneous in terms of body and gWAT weights, some of them showing weights closed to CD mice and others being clearly overweight and fat. However, both F3-WD5 populations were significantly heavier and fatter (p<0.001 and p<0.01, respectively) than control and F3-WD1 populations (Figure 3B-I and Figure 3—source data 1,2).

Lines 275-282

The main limitation of our study is the phenotypic heterogeneity observed in the males of the WD4 and WD5 generations (Figure 1—figure supplement 1C) and in the CD-fed WD5 progenies (Figure 3) which might lead to biased conclusions. Indeed, although the statistical tests we used here should rule out this weakness, we cannot rule out the possibility of the presence of sub-populations. Metabolic heterogeneity induced by an unbalanced diet has already been reported in mice (10, 11) and the strong heterogeneity observed in our model may indicate an adaptative process whereby different subpopulations could emerge in response to the maintenance of an unbalanced diet.

References:

1. Zhou Y, Zhu H, Wu HY, Jin LY, Chen B, Pang HY, et al. Diet-Induced Paternal Obesity Impairs Cognitive Function in Offspring by Mediating Epigenetic Modifications in Spermatozoa. Obesity (Silver Spring). 2018;26(11):1749-57.

2. Massiera F, Barbry P, Guesnet P, Joly A, Luquet S, Moreilhon-Brest C, et al. A Western-like fat diet is sufficient to induce a gradual enhancement in fat mass over generations. J Lipid Res. 2010;51(8):2352-61.

3. Fullston T, Ohlsson Teague EM, Palmer NO, DeBlasio MJ, Mitchell M, Corbett M, et al. Paternal obesity initiates metabolic disturbances in two generations of mice with incomplete penetrance to the F2 generation and alters the transcriptional profile of testis and sperm microRNA content. Faseb j. 2013;27(10):4226-43.

4. Cropley JE, Eaton SA, Aiken A, Young PE, Giannoulatou E, Ho JW, et al. Male-lineage transmission of an acquired metabolic phenotype induced by grand-paternal obesity. Molecular metabolism. 2016;5(8):699-708.

5. Benjamini Y, Hochberg Y. Controlling the false discovery rate: A practical and powerful approach to multiple testing. Journal of the Royal Statistical Society,. 1995;57(1):289-300.

6. Horan TS, Marre A, Hassold T, Lawson C, Hunt PA. Germline and reproductive tract effects intensify in male mice with successive generations of estrogenic exposure. PLoS Genet. 2017;13(7):e1006885.

7. Masuyama H, Mitsui T, Eguchi T, Tamada S, Hiramatsu Y. The effects of paternal high-fat diet exposure on offspring metabolism with epigenetic changes in the mouse adiponectin and leptin gene promoters. Am J Physiol Endocrinol Metab. 2016;311(1):E236-45.

8. Nätt D, Kugelberg U, Casas E, Nedstrand E, Zalavary S, Henriksson P, et al. Human sperm displays rapid responses to diet. PLoS Biol. 2019;17(12):e3000559.

9. Zhang Y, Zhang X, Shi J, Tuorto F, Li X, Liu Y, et al. Dnmt2 mediates intergenerational transmission of paternally acquired metabolic disorders through sperm small non-coding RNAs. Nat Cell Biol. 2018;20(5):535-40.

10. Burcelin R, Crivelli V, Dacosta A, Roy-Tirelli A, Thorens B. Heterogeneous metabolic adaptation of C57BL/6J mice to high-fat diet. Am J Physiol Endocrinol Metab. 2002;282(4):E834-42.11. Dumas ME, Rothwell AR, Hoyles L, Aranias T, Chilloux J, Calderari S, et al. Microbial-Host Co-metabolites Are Prodromal Markers Predicting Phenotypic Heterogeneity in Behavior, Obesity, and Impaired Glucose Tolerance. Cell Rep. 2017;20(1):136-48.